# Empirical Evaluation of Unoptimized Sorting Algorithms on 8-Bit AVR Arduino Microcontrollers

**DOI:** 10.3390/s26010214

**Published:** 2025-12-29

**Authors:** Julia Golonka, Filip Krużel

**Affiliations:** Faculty of Computer Science and Mathematics, Cracow University of Technology, 31-155 Kraków, Poland; julia.golonka@student.pk.edu.pl

**Keywords:** embedded systems, 8-bit AVR microcontrollers, Arduino boards, sorting algorithms, performance evaluation, execution time, SRAM usage, write operations, resource-constrained devices

## Abstract

Resource-constrained sensor nodes in Internet-of-Things (IoT) and embedded sensing applications frequently rely on low-cost microcontrollers, where even basic algorithmic choices directly impact latency, energy consumption, and memory footprint. This study evaluates six sorting algorithms—Bubble Sort, Insertion Sort, Selection Sort, Merge Sort, Quick Sort, and Heap Sort—in the restricted environment that microcontrollers provide. Three Arduino boards were used: Arduino Uno, Arduino Leonardo, and Arduino Mega 2560. Each algorithm was implemented in its unoptimized form and tested on datasets of increasing size, emulating buffered time-series sensor readings in random, ascending, and descending order. Execution time, number of write operations, and memory usage were measured. The tests show clear distinctions between the slower $O(n^{2})$ algorithms and the more efficient O(nlogn) algorithms. For random inputs of n=1000 elements, Bubble Sort required 1,958,193.75 μson average, whereas Quick Sort completed it in 54,260.50 μs and Heap Sort in 92,429.00 μs, i.e., speedups of more than one order of magnitude compared to the quadratic baseline. These gains, however, come with very different memory footprints. Merge Sort kept the runtime below 100,000 μs at n=1000 but required approximately 2023 bytes of additional static random-access memory (SRAM), effectively exhausting the 2 kB SRAM of the Arduino Uno. QuickSort used approximately 311 bytes of extra SRAM and failed to process larger ascending and descending datasets on the more constrained boards due to its recursive pattern and stack usage. Heap Sort offered the best overall trade-off: it successfully executed all tested sizes up to the SRAM limit of each board while using only about 12–13 bytes of additional SRAM and keeping the runtime below 100,000 μs for n=1000. The results provide practical guidelines for selecting sorting algorithms on 8-bit AVR Arduino-class microcontrollers, which are widely used as simple sensing and prototyping nodes operating under strict RAM, program-memory, and energy constraints.

## 1. Introduction

### 1.1. Aim and Scope

The primary aim of this study is to compare the behavior of several classic sorting algorithms when executed on 8-bit AVR Arduino-class microcontroller boards with severe resource constraints. The tested algorithms include Bubble Sort, Insertion Sort, Selection Sort, Merge Sort, Quick Sort, and Heap Sort, representing both $O(n^{2})$ and O(nlogn) time complexity classes. The algorithms were evaluated on three widely used Arduino boards: Arduino Uno, Arduino Leonardo, and Arduino Mega 2560. Each board’s processor runs at a 16 MHz clock, but they differ significantly in static random-access memory (SRAM) capacity and program memory, which directly influence the maximum problem size and the viability of certain algorithmic strategies.

Each algorithm was tested using arrays of increasing sizes and various element orders: random, ascending, and descending. To ensure fairness and to observe their natural behavior in a straightforward implementation scenario, no algorithm-specific low-level optimizations were applied. The results aim to provide insight into the suitability of each algorithm for constrained environments and to offer practical guidelines for selecting sorting algorithms for 8-bit AVR Arduino-class sensing nodes that act as simple prototypes for IoT deployments.

To make these objectives more precise, we formulate the following research questions:**RQ1:**  For typical 8-bit AVR Arduino-class boards, what is the largest array size that can be reliably sorted by each classical algorithm under strict SRAM constraints?**RQ2:**  How do execution time, number of array-element writes, and SRAM usage trade off between $O(n^{2})$ algorithms and O(nlogn) algorithms on such platforms?**RQ3:**  Which of the considered algorithms offers the most robust default choice for managing queues and buffered sensor readings on low-end Arduino-class microcontrollers?

### 1.2. Justification of the Topic Selection

Sorting algorithms are a fundamental concept in computer science and are typically among the first topics introduced to new learners. Despite their apparent simplicity and intuitive nature, their performance can vary significantly depending on both the software implementation and the underlying hardware. Classical analyses, however, usually assume either an abstract machine model or desktop/server-class processors with abundant memory and deep cache hierarchies. As a result, there is comparatively less empirical data on how textbook algorithms behave on devices with only a few kilobytes of RAM and relatively low clock frequencies, such as those commonly used in embedded sensing platforms.

The Arduino ecosystem was specifically chosen as the testing environment due to its popularity, accessibility, and representative nature for educational, prototyping, and even certain production-grade sensing applications. Its limited resources make it ideal for testing algorithm performance under constraints that differ substantially from those of typical computers. Moreover, the ubiquity of Arduino boards and the availability of open-source toolchains facilitate the reproducibility of experiments and the adoption of the presented methodology. The results of this work can therefore serve not only as a guide for developers when selecting algorithms for small systems and optimizing their performance, but also as an educational reference illustrating how algorithmic complexity interacts with real-world hardware constraints.

### 1.3. Relevance to IoT and Edge Devices

The constraints inherent to Arduino boards are highly relevant for Internet of Things (IoT) and edge computing nodes, since these devices typically operate with limited memory, processing power, and energy budgets. Many sensor nodes and embedded data-acquisition systems are built around 8-bit or 32-bit microcontrollers with only a few kilobytes of SRAM, yet they are expected to perform non-trivial local data processing to reduce communication overhead and extend battery lifetime.

Sorting operations are utilized in various tasks, including buffering and ordering time-series sensor readings, implementing median and rank-based filters, prioritizing events in queues, and preparing data for local aggregation or compression before transmission. The choice of sorting algorithm, therefore, affects not only latency but also energy consumption, memory wear (in the case of flash-backed storage), and the overall quality of service in a sensing network. By systematically evaluating unmodified sorting algorithms on representative Arduino-class hardware, this study provides insight into which algorithms remain feasible as dataset sizes grow and which quickly become impractical due to their memory footprint or execution time. These observations are directly applicable to the design of IoT and edge platforms where on-node pre-processing is necessary but must be balanced against strict resource limitations.

It is essential to note that many contemporary IoT and edge deployments are based on 32-bit microcontrollers with substantially larger SRAM and flash memory, and in many cases, also with hardware floating-point support, as reflected in recent embedded benchmark suites and industrial reports [1,2]. In this work, we deliberately focus on low-end 8-bit AVR Arduino-class boards, which remain widely used in education, rapid prototyping, and simple sensing nodes. Extending the present comparison to representative 32-bit Cortex-M and digital signal processing (DSP)-class microcontrollers, including floating-point workloads, is left as future work.

### 1.4. Related Work

The performance of sorting algorithms on constrained platforms has been studied from several complementary perspectives. A substantial line of work focuses on optimizing sorting for devices with very limited RAM but relatively large non-volatile storage (e.g., flash memory). Jackson, Gresl, and Lawrence proposed a no-output-buffer variant of external merge sort tailored for embedded devices with less than 4 KB of available RAM, reducing I/O operations and achieving up to 35% lower execution time compared to baseline approaches [3]. Jackson and Lawrence later extended this design with an algorithm explicitly optimized for flash memory layouts, which further improved execution time by roughly 30% [4]. Lawrence subsequently introduced an adaptive flash sorting strategy that selects between read-based and merge-based methods at runtime to accommodate changing workload characteristics [5]. Related flash-oriented techniques include FSort, which targets sensor nodes equipped with flash memory and focuses on minimizing erase cycles and page writes [6]. Follow-up work on fast external sorting for flash-based sensor devices and embedded platforms has also been conducted [7,8].

Beyond these flash-specific methods, a broader body of research exists on evaluating algorithm performance on microcontrollers and embedded systems. MiBench is a widely used benchmark suite that defines representative embedded workloads and measurement procedures for constrained platforms [1]. CoreMark, developed by EEMBC, has become a de facto standard for characterizing the performance of microcontrollers, emphasizing simple, reproducible timing and memory measurements [2]. More specialized studies, such as the sliding-window-based benchmarking of embedded microcontrollers presented by Nakutis, investigate how microcontroller architecture and memory organization affect execution time and resource usage under realistic workloads [9]. In parallel, recent comparative studies have examined sorting algorithms implemented on FPGAs or more powerful processors to explore hardware acceleration and architectural trade-offs [10,11].

Experimental comparisons of classical sorting algorithms on general-purpose computers have also been reported in the literature. These works typically measure execution time (and sometimes memory usage) for canonical algorithms such as Bubble Sort, Insertion Sort, Selection Sort, Merge Sort, Quick Sort, and Heap Sort across a range of input sizes and data characteristics [12,13,14]. However, they are usually conducted on desktop, server, or high-performance platforms with abundant RAM and deep cache hierarchies, where memory capacity is rarely the limiting factor.

In contrast to these prior efforts, the present study deliberately evaluates unmodified, textbook implementations of six well-known sorting algorithms on low-end 8-bit AVR-based Arduino boards. Rather than proposing new external-memory or hardware-accelerated techniques, we aim to establish a reproducible baseline for how these algorithms behave under strict SRAM constraints that are typical for low-cost sensing nodes. The proposed metrics—execution time, number of array-element writes, and measured SRAM usage—complement existing embedded benchmark suites and provide practitioners with guidance for selecting sorting algorithms for simple, RAM-limited microcontroller-based systems.

### 1.5. Contributions and Paper Organization

This work makes the following contributions:It provides a systematic experimental comparison of six classical, textbook sorting algorithms—Bubble Sort, Insertion Sort, Selection Sort, Merge Sort, Quick Sort, and Heap Sort—implemented in an unoptimized form on three representative 8-bit AVR Arduino-class microcontrollers.It quantifies the combined impact of execution time, number of array-element writes, and measured SRAM usage across different input sizes and data orderings, highlighting the practical limits at which individual algorithms cease to be viable on RAM-constrained boards such as the Arduino Uno.It shows that, within the examined setting, Heap Sort offers the most robust default choices for managing buffered sensor readings and queue-like workloads on 8-bit AVR Arduino-class microcontrollers, successfully handling all tested problem sizes while using only a small, constant amount of additional SRAM.

The remainder of this article is organized as follows. Section 2 (*Experimental Setup and Methodology*) describes the hardware platforms, the selected algorithms, and the measurement procedures used in this study. Section 3 presents detailed experimental results for execution time, write operations, and memory usage, both per board and in aggregated form. Section 4 discusses the implications of these results for the design of low-end microcontroller-based sensing nodes and outlines the limitations of the present work. Finally, Section 5 summarizes the main findings and indicates directions for future research.

## 2. Experimental Setup and Methodology

### 2.1. Description of the Examined Microcontrollers

This study uses three popular Arduino-class development boards: Arduino Uno, Arduino Leonardo, and Arduino Mega 2560. All three are based on 8-bit AVR microcontrollers running at a 16 MHz clock, but they differ significantly in their on-chip memory resources:Arduino Uno—based on the ATmega328P microcontroller, which provides 32 KB of flash program memory, 2 KB of SRAM, and 1 KB of EEPROM [15,16]. It is one of the most widely used boards for introductory embedded-system projects and small-scale sensing applications.Arduino Leonardo—uses the ATmega32U4 microcontroller, which also offers 32 KB of flash and 1 KB of EEPROM, but increases the available SRAM to 2.5 KB [17,18]. This additional SRAM allows it to handle slightly larger datasets and more complex sketches than the Uno, while remaining in the same low-end 8-bit AVR family.Arduino Mega 2560—the most memory-rich board in this comparison. Its ATmega2560 microcontroller provides 256 KB of flash, 8 KB of SRAM, and 4 KB of EEPROM [19,20]. The substantially larger SRAM capacity enables the execution of algorithms on much larger in-memory datasets than on the Uno and Leonardo.

All three boards operate at 5 V and share the same 8-bit AVR instruction set without hardware floating-point units. Consequently, all experiments in this study are carried out with integer (fixed-point) arithmetic, which matches typical usage patterns on these platforms.

### 2.2. Description of the Selected Algorithms

In this subsection, we briefly review the six classical sorting algorithms that constitute the focus of our experimental study: Bubble Sort, Insertion Sort, Selection Sort, Merge Sort, Quick Sort, and Heap Sort. Together, they cover both quadratic-time ($O(n^{2})$) and O(nlogn) algorithms that are widely used as pedagogical examples and baseline methods in textbooks and software libraries. The selection intentionally excludes heavily optimized or hardware-specialized variants so that the results capture the behavior of straightforward, “textbook-style” implementations on 8-bit AVR Arduino-class microcontrollers.

For each algorithm, we provide an intuitive description, a small illustrative example, and pseudocode in an array based in-place formulation (Algorithms 1–9). These pseudocode listings follow standard treatments that can be found in classical algorithm texts, for example [21,22,23], adapted only to match the C++ implementation style used on the microcontrollers.

A summary of the theoretical properties of the six algorithms—time complexity, auxiliary memory requirements, and stability—is given in Table 1. The subsequent experimental sections relate these well-known properties to the actual behavior observed on Arduino-class hardware under strict SRAM constraints.

#### 2.2.1. Bubble Sort

Bubble Sort works by iterating through the list multiple times and comparing neighboring elements. It swaps them if they are in the wrong order until the list is sorted.

The Algorithm 1 iterates through the list of *n* elements, selecting the largest one and bringing it to the last position. The algorithm requires n−1 iterations, where each subsequent pass performs one fewer comparison than the previous one. The implemented version of the algorithm terminates the iteration early if no swaps occur, allowing it to finish immediately when the array is already sorted.
**Algorithm 1:** Bubble Sort
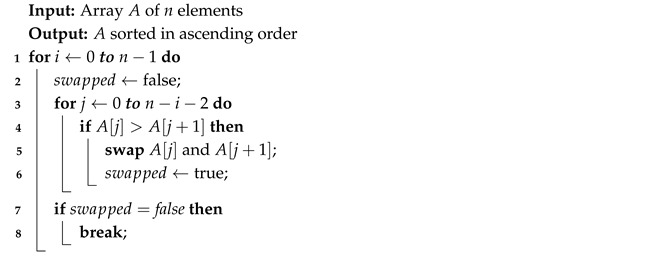


**Example** With the list A=[5,2,4,1,3].

**Iteration 1:** 
[5,2,4,1,3]→[2,5,4,1,3]→[2,4,5,1,3]→[2,4,1,5,3]→[2,4,1,3,5]The largest element (5) was moved to the last position.**Iteration 2:** 
[2,4,1,3,5]→[2,1,4,3,5]→[2,1,3,4,5]The second largest element (4) was moved to the second-last position.**Iteration 3:** 
[2,1,3,4,5]→[1,2,3,4,5]The third largest element (3) was moved to the third-to-last position. At this point, the list is fully sorted.

##### Complexity and Stability

The worst-case complexity of Bubble Sort is identical to the average-case and equals $O(n^{2})$. The worst-case occurs when the list is completely reversed. The best-case complexity occurs when the list is already sorted and equals O(n). The algorithm is stable, meaning that the equal-value elements stay in their relative positions.  

##### Suitability

With its high inefficiency, Bubble Sort’s only advantage is simplicity. It is primarily used for educational demonstrations or for sorting small lists, where the $O(n^{2})$ complexity does not result in noticeable delays.

#### 2.2.2. Insertion Sort

Insertion Sort works by dividing the list into a sorted and unsorted part, then taking each element from the unsorted and inserting it in a correct position in the sorted section.

The Algorithm 2 starts at the second element of a list of *n* elements. It compares the value to the previous elements until it finds the correct position. It requires n−1 iterations.

**Example** With the list A=[5,2,4,1,3].

**Iteration 1:** 
[5,2,4,1,3]→[2,5,4,1,3]The first (5) and the second (2) element are now sorted.**Iteration 2:** 
[2,5,4,1,3]→[2,4,5,1,3]The third element (4) entered the correct position.**Iteration 3:** 
[2,4,5,1,3]→[2,4,1,5,3]→[2,1,4,5,3]→[1,2,4,5,3]The fourth element (1) entered the correct position.**Iteration 4:** 
[1,2,4,5,3]→[1,2,4,3,5]→[1,2,3,4,5]The last element (3) entered the correct position. The list is now fully sorted.

##### Complexity and Stability

Its complexity is the same as that of the Bubble Sort, with the average and worst case being $O(n^{2})$, and the best case being O(n). These occur, respectively, when the list is in reverse order and when it is already sorted. The algorithm is stable.  

##### Suitability

With a complexity of the same order as Bubble Sort, its purpose is likewise limited to small or nearly sorted lists. However, unlike Bubble Sort, Insertion Sort is also employed as a component of hybrid algorithms, where its efficiency on small inputs is advantageous.
**Algorithm 2:** Insertion Sort
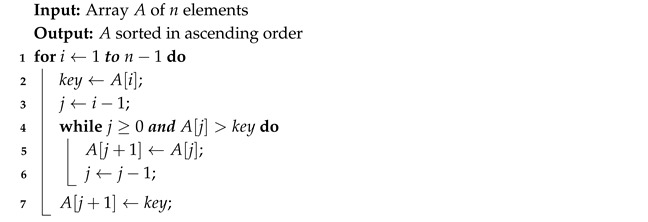


#### 2.2.3. Selection Sort

Like the Insertion Sort, the Selection Sort divides the list into two. It finds the minimum value element from the unsorted part and moves it to the end of the sorted part.

Algorithm 3 initially starts at the first element of a list of *n* elements. It iterates through the unsorted portion of the list, finds the smallest element, and swaps it with the first element of the unsorted section. The swapped element becomes part of the sorted section and is skipped in subsequent iterations. This process is repeated n−1 times until the entire list is sorted.
**Algorithm 3:** Selection Sort
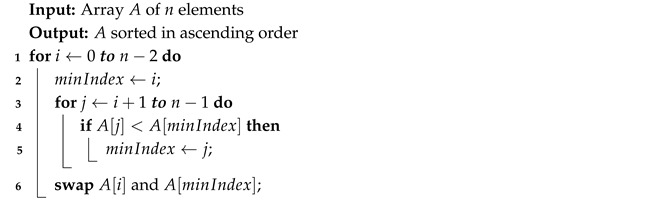


**Example** With the list A=[5,2,4,1,3].

**Iteration 1:** 
[5,2,4,1,3]→[1,2,4,5,3]The smallest item (1) was found and brought to the front.**Iteration 2:** 
[1,2,4,5,3]The second smallest item (2) was found. It was already in the right position.**Iteration 3:** 
[1,2,4,5,3]→[1,2,3,5,4]The third smallest item (3) was found and moved to the third position.**Iteration 4:** 
[1,2,3,5,4]→[1,2,3,4,5]The fourth smallest item (4) was found and moved to the fourth position. The list is sorted.

##### Complexity and Stability

In Selection Sort, the initial order of elements does not affect its behavior. The worst, average, and best cases all have a time complexity of $O(n^{2})$. Additionally, because the algorithm performs swaps, it is not stable.  

##### Suitability

Due to its straightforward logic, Selection Sort is well-suited for educational purposes and teaching sorting fundamentals. Its practical use is generally limited to small lists or memory-constrained environments.

#### 2.2.4. Merge Sort

Merge Sort recursively splits the list into halves, then it merges the sublists back together in ascending order.

The Algorithm 4 recursively divides a list of *n* elements into halves. When each sublist contains only one element, adjacent sublists (Algorithm 5) are merged back together in ascending order.

**Example** With the list A=[5,2,4,1,3].

**Divide:** 
[5,2,4,1,3]→[5,2],[4,1,3]→[5],[2],[4],[1,3]→[5],[2],[4],[1],[3]**Merge:** 
[5],[2],[4],[1],[3]→[2,5],[4],[1,3]→[2,5],[1,3,4]→[1,2,3,4,5]

##### Complexity and Stability

The algorithm is stable and has the worst, average, and best case complexity of O(nlogn).  

##### Suitability

Merge Sort performs efficiently on large lists due to its guaranteed O(nlogn) complexity. However, its main drawback is additional memory usage for merging, making it less suitable for memory-constrained environments.   
**Algorithm 4:** Merge Sort
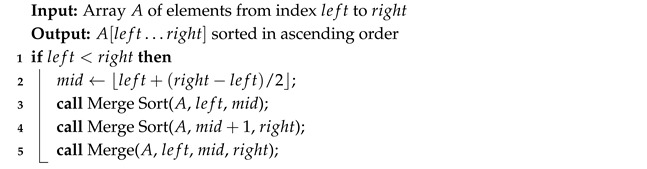


**Algorithm 5:** Merge


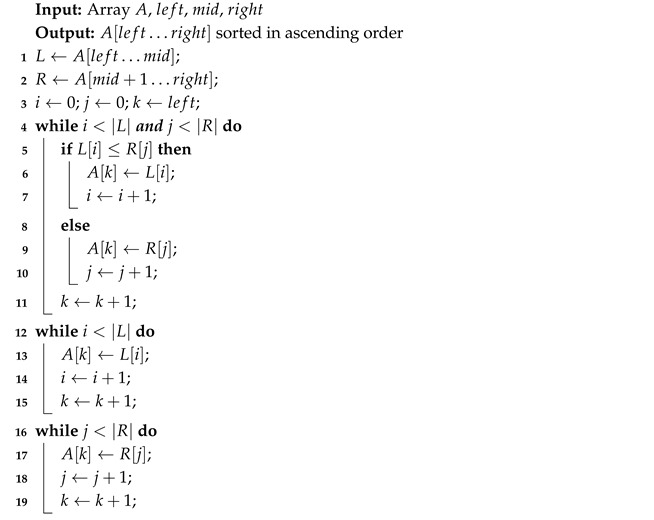




#### 2.2.5. Quick Sort

Each iteration of Quick Sort (Algorithm 6) selects a pivot that then acts as a divider. The list then splits into smaller and larger elements than the pivot.

Algorithm 7 chooses a pivot from the list of *n* elements. Then it moves all the smaller elements to its left side and the larger to the right. Then, it recursively does the same on both new sublists.
**Algorithm 6:** Quick Sort
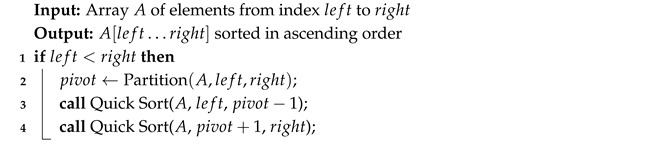


**Algorithm 7:** Partition


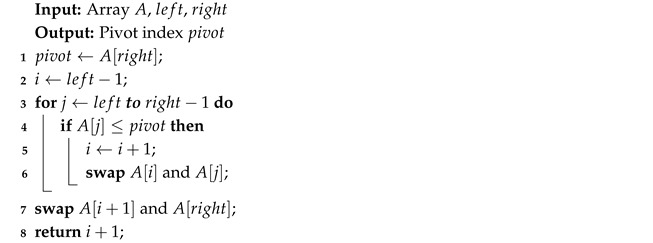




**Example** With the list A=[5,2,4,1,3].

**Partition 1:** 
[5,2,4,1,3]→[2,1,3,5,4]The last element was the pivot (3), and now the elements smaller than 3 are to its left and the larger to its right.**Partition 2:** 
[2,1]→[1,2]
[5,4]→[4,5]Both left and right elements are now sorted (with 1 and 4, respectively, being pivots). The list is sorted.

##### Complexity and Stability

Quick sort is not stable. The elements equal to the pivot could end up reordered when partitioning. Its best and average-case complexities are O(nlogn), but its worst-case complexity can be up to $O(n^{2})$, when the pivot produces largely uneven sublists.  

##### Suitability

Due to its efficiency, Quick Sort is suitable for large lists and is often preferred when average performance and low memory usage are important.

#### 2.2.6. Heap Sort

Heap Sort relies on building a binary heap from the list. Then, it takes the largest element and puts it at the end of the list.

Algorithm 8 starts by constructing a max heap from the list of *n* elements. Then, swapping the root with the last element effectively removes it from the heap. Afterwards, the new root is heapified (Algorithm 9) to restore the structure.
**Algorithm 8:** Heap Sort
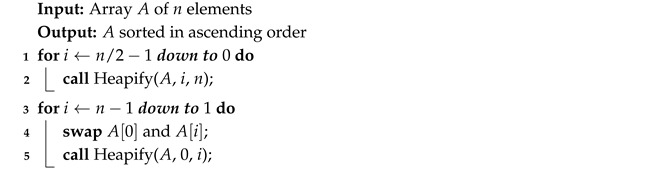


**Algorithm 9:** Heapify


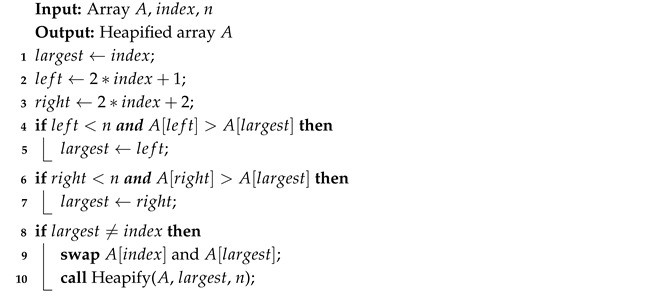




**Example** With the list A=[5,2,4,1,3].

**Initial Max Heap:** 
[5,2,4,1,3]→[5,3,4,1,2]**Iteration 1:** 
[5,3,4,1,2]→[2,3,4,1,5]→[4,3,2,1,5]The root element (5) is swapped with the last element of the heap, and the root is re-heapified.**Iteration 2:** 
[4,3,2,1,5]→[1,3,2,4,5]→[3,1,2,4,5]The root element (4) is swapped with the last element of the heap, and the root is re-heapified.**Iteration 3:** 
[3,1,2,4,5]→[2,1,3,4,5]The root element (3) is swapped with the last element of the heap. No re-heapification is required.**Iteration 4:** 
[2,1,3,4,5]→[1,2,3,4,5]The root element (2) is swapped with the last element of the heap. The array is sorted.

##### Complexity and Stability

In the best, average, and worst cases, the Heap Sort has a complexity of O(nlogn). It is not stable.  

##### Suitability

Heap sort is highly efficient and, unlike Merge Sort, does not require additional memory, making it the top choice of this selection. It is suitable for large lists; however, it is less commonly used than Quick Sort due to its less efficient cache utilization.

The key theoretical properties of the six algorithms discussed above are summarized in Table 1. This includes their asymptotic time complexity in the best, average, and worst cases, the amount of auxiliary memory required beyond the input array, and whether the algorithm is stable.

### 2.3. Research Methodology

#### 2.3.1. Choice of Programming Language

Since Arduino IDE provides C++, it was used to write the entire code. It is a standard for Arduino development, offering direct access to hardware. It also ensures efficient performance and memory usage.  

#### 2.3.2. Choice of Development Environment

Arduino IDE version 2.3.6 was used. It is compatible with all Arduino boards examined in this study and provides an integrated toolchain for compiling and uploading sketches. The code was compiled using the default AVR-GCC-based toolchain shipped with this IDE, with size-oriented optimization enabled (equivalent to the -Osoptimization level). After compilation and uploading, the boards were connected to the host PC via a USB cable and executed the sorting routines as standalone kernels; no Serial I/O or other user-level tasks were performed during the timed region (standard timer interrupts required by micros() remained enabled).  

#### 2.3.3. Choice of Hardware

In terms of hardware, three different boards were selected: Arduino Uno, Arduino Leonardo, and Arduino Mega 2560. They differ primarily in available SRAM and I/O capabilities, while sharing the same 16 MHz 8-bit AVR architecture. Together, they form a realistic and low-cost cross-section of Arduino-class hardware used in educational and prototyping scenarios.


**Arduino Uno**
Since the Arduino Uno is likely the first board that every beginner Arduino developer buys or hears about, it was an obvious choice. It should illustrate an average reference point for performance in typical Arduino programs.
**Arduino Leonardo**
The Arduino Leonardo offers a 25% increase in SRAM compared to the Arduino Uno. It was included in the study to evaluate whether even such a modest upgrade in memory capacity has a noticeable impact on sorting algorithm performance.
**Arduino Mega 2560**
The Arduino Mega 2560 is the most capable board selected for this comparison. With 400% more SRAM than the Arduino Uno, it serves as a high-end benchmark for evaluating how algorithms behave under significantly larger memory capacity.

Choosing three boards from the same microcontroller family but with markedly different SRAM sizes enables us to study how memory constraints alone affect algorithm viability, while keeping other factors, such as clock speed, SRAM access time, and instruction set, effectively constant.  

#### 2.3.4. Implemented Algorithms

The following algorithms were implemented in their unoptimized form to avoid introducing bias. The goal was to compare their inherent logic and behavior rather than fine-tuned performance, as real-world data is rarely ideal.


**Bubble Sort**
Although inefficient, Bubble Sort serves as a baseline due to its understandability. It provides a good comparison point, so it was a necessity in this comparison.
**Insertion Sort**
Insertion Sort was chosen as it performs efficiently on small or partially sorted datasets. Its implementation allows comparison with Bubble Sort as another simple, quadratic-time algorithm.
**Selection Sort**
Selection Sort was included for its simplicity and predictable number of comparisons and swaps. It serves as another reference among the $O(n^{2})$ algorithms.
**Merge Sort**
Merge Sort was implemented to represent a divide-and-conquer algorithm with a time complexity of O(nlogn). It provides insight into how more advanced algorithms perform on memory-constrained microcontrollers.
**Quick Sort**
Quick Sort was implemented as another divide-and-conquer algorithm. Known for its efficiency in practice, it allows comparison with Merge Sort and the quadratic algorithms.
**Heap Sort**
Heap Sort was included as a comparison of an O(nlogn) algorithm that relies on a binary heap. Unlike Merge Sort, it sorts in place without requiring additional memory, making it especially relevant for memory-constrained microcontrollers. Its implementation allows evaluation of performance trade-offs between in-place and memory-intensive sorting methods.

#### 2.3.5. Implementation Details

No libraries were used, and all algorithms were implemented in C++, allowing full control over memory usage and the number of write operations. The implementations were based on translating pseudocode into Arduino-compatible C++ code and were verified through test runs to ensure correctness.  

#### 2.3.6. Write Count Tracking

Each algorithm counted array-element write operations by incrementing a counter after each swap or movement of an element in the main array. Writes to other variables, including temporaries and loop counters, were not included, as the goal was to approximate the volume of data traffic to the array itself rather than to build a full energy or memory-wear model.  

#### 2.3.7. Memory Tracking

A function (Algorithm 10) to compute the free SRAM was called before, after, and multiple times during the execution of each algorithm, including within loops and recursive calls. The smallest observed value of free RAM was recorded, and the memory usage was calculated by subtracting this value from the amount of SRAM available before the algorithm started. For the Memory Usage Tracking implementations, the following variables were used:__heap_start: the address of the heap beginning, where memory allocation starts.__brkval: the address of the heap end, the top of the memory allocated. If the heap has not grown yet, __brkval is 0.The address of a local variable (v): represents the current top of the stack. The stack grows from high to low addresses, so a higher address corresponds to lower memory usage.

#### 2.3.8. Validation of Memory Tracking Method

To verify the correctness of the SRAM tracking approach, a controlled experiment was conducted using a recursive function that allocates a known number of bytes on the stack at each recursion depth.   
**Algorithm 10:** Compute Free SRAM
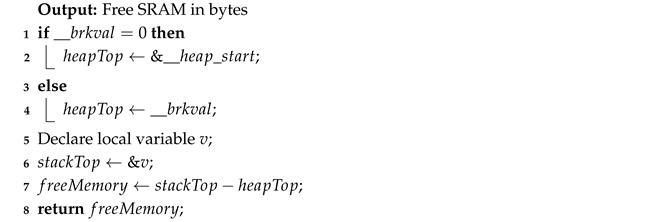


#### 2.3.9. Memory Usage Validation

The experimental results, summarized in Table 2, demonstrate a clear correlation between the allocated memory size and the decrease in free SRAM. The observed memory usage increases proportionally with both recursion depth and local array size, closely matching theoretical expectations, with a small, consistent offset. This offset corresponds to the fixed memory overhead associated with function call frames, compiler alignment, and bookkeeping in the heap. The consistent and predictable behavior across varying parameters confirms that the implemented (Algorithm 10) method reliably reflects real-time SRAM consumption. It should be noted that this validation was performed under the same conditions as the main experiments, i.e., without dynamic heap allocation, except for the statically defined data structures, and with a conventional downward-growing stack. Under these assumptions, the freeMemory() method provides a reliable proxy for the currently available SRAM and for the additional memory consumed by each sorting routine. For different microcontroller architectures, runtime libraries, or memory-allocation schemes, the calibration procedure illustrated in Table 2 would need to be repeated before interpreting the absolute values of SRAM usage. This experiment confirmed that the method correctly reports stack and SRAM usage up to at least the maximum recursion depths observed in the sorting benchmarks.

#### 2.3.10. Time Measurement

Arduino provides the micros() function, which returns the number of microseconds elapsed since the program started. It was called immediately before and after each sorting routine, and the start value was subtracted from the end value to obtain the total execution time. On the 16 MHz AVR boards used in this work, micros() advances in 4 μs steps, so measurements for very short runs (small *N* and already-sorted inputs) are quantized. In the sorting experiments, however, most execution times were on the order of tens of thousands to several million microseconds, i.e., several orders of magnitude larger than the 4 μs resolution. Consequently, the quantization error introduced by the timer had a negligible impact on the reported means and standard deviations. Only for the smallest array sizes do some measurements collapse to identical values due to this resolution; these points are still shown for completeness, but the interpretation of performance trends is based primarily on the larger array sizes, where the effective timer resolution is fully adequate. No Serial I/O or other user-level tasks were performed during the timed region; standard timer interrupts required by micros() remained enabled. Although the array sizes (*N*) are discrete variables, the results in the figures are connected by lines to clearly visualize the empirical scaling trend (consistent with expected complexity) of the algorithms.  

#### 2.3.11. Output Validation

To verify the correctness of each implemented algorithm, two validation methods were applied for every experiment: checksum comparison and order verification. First, the sum of all elements was computed both before and after sorting; matching checksums ensured that no elements were lost or spuriously introduced. Second, after sorting, the program was iterated through the array once to verify that every element was less than or equal to its successor. Both checks were performed for all array sizes, input orders, and boards, and no mismatches were observed, which gives high confidence that the implementations are functionally correct.  

#### 2.3.12. Data Types

The dataset elements were all signed 16-bit integers, matching the native integer type of the tested AVR microcontrollers. This reflects typical usage on Arduino-class boards, where sensor readings, counters, and identifiers are often stored as integers, allowing for a clean comparison of algorithmic behavior without introducing additional variability from floating-point emulation. Floating-point data are crucial in many DSP workloads, but their processing efficiency strongly depends on the quality of the underlying floating-point support, which is therefore left for future work outside the scope of this study.  

#### 2.3.13. Array Sizes

The algorithms were tested using a variety of dataset sizes. For each board, arrays of 100, 250, 500, 750, and 1000 elements were used. However, due to memory limitations, some algorithm–board combinations were unable to handle the largest datasets. In such cases, the corresponding entries in the tables are marked with “–”, and these missing runs are excluded from the averages reported in the aggregated figures.  

#### 2.3.14. Element Generation

Three different orders of elements were tested: random, ascending, and descending.


**Random Order**
The most important of the three, in random order, represents a set that most closely reflects typical algorithm usage. Each element was generated within a loop by a random() function. Each element’s integer value ranged from 0 to 10,000, and duplicate values were allowed.
**Ascending Order**
The ascending ordered dataset was tested for efficiency evaluation on an array that, theoretically, requires no modifications. It was included in the experiments to observe algorithm behavior under ideal conditions rather than to represent a real-world scenario, since such perfectly ordered data rarely occurs in practice. This set contains elements that have their index as their value (array[i] = i).
**Descending Order**
The typical worst-case scenario is also largely theoretical; however, it is significantly more important to consider. In this case, it is not only a matter of unusually fast execution but rather a potentially program-breaking situation. This dataset consists of elements with values determined by subtracting their index and one from the total array size (array[i] = ARRAY_SIZE - i - 1).

#### 2.3.15. Repetition

For each board and algorithm, the randomly ordered dataset was tested most extensively with eight independent repetitions for every array size. Since both the ascending and descending order datasets were expected not to produce highly variable results, they were tested three times each. For every configuration (board, algorithm, input order, and array size), we report the arithmetic mean of the measured execution times across repetitions.

The same set of repetitions was also used to quantify measurement variability. For each individual board and configuration, the standard deviation (SD) of execution time across repeated runs was computed using the built-in Excel function. In figures where results are aggregated across all three boards, the SD value shown as an error bar corresponds to the arithmetic mean of the per-board SD values and is reported as a descriptive indicator of measurement variability (not as a formal pooled estimator). Thus, all error bars in the figures represent ±1 standard deviation around the corresponding mean execution time. For many configurations, the standard deviation was very small compared to the mean execution time, so the error bars are narrower than the marker size and may appear visually indistinguishable at the scale of the figure. 

#### 2.3.16. Outlier Handling

We did not apply any outlier removal. All repeated runs were included in the reported means and standard deviations. The only exclusion criterion would be a failed correctness check (checksum or non-decreasing order verification), which did not occur in any run. 

#### 2.3.17. Output Format

All results were output as simple numbers separated by tabs. Each run recorded the array size, runtime in microseconds, free memory before the execution, lowest free memory observed during execution, and the number of write operations. The data was then imported into Excel, where the results were averaged to obtain a single value for each board, algorithm, and dataset type. From there, another average was calculated across the three Arduino boards to consolidate the three tables into a single one.

The overall structure of the experiments follows common practice in embedded benchmarking, where relatively small, deterministic kernels are executed repeatedly under controlled conditions and evaluated in terms of execution time and memory footprint. This philosophy is similar in spirit to established benchmark suites, such as MiBench and CoreMark, which provide representative workloads and measurement procedures for embedded processors [1,2]. The idea of systematically varying array size and data ordering across repeated runs is also conceptually related to sliding-window style microcontroller benchmarks, such as those proposed by Nakutis for evaluating how architectural and memory-organization details affect execution behavior [9].

## 3. Research Results

### 3.1. Performance per Microcontroller Board

#### 3.1.1. Bubble Sort

Table 3 shows the execution times of Bubble Sort across the three Arduino boards for different input sizes and order types. Although the execution times are similar due to the identical 16 MHz clock speed across all boards, the key differentiator is the completion capability. The larger SRAM of the Mega 2560 enables the execution of memory-intensive algorithms (such as Merge Sort and Quick Sort) on larger datasets, whereas the Uno fails, highlighting the critical role of memory capacity over raw processing speed in this class of devices. While slight variations in execution time exist between boards due to minor architectural differences, these are negligible compared to the impact of memory constraints.

#### 3.1.2. Insertion Sort

Table 4 shows results similar to those in Table 3, showing no significant differences in execution times for Insertion Sort.

#### 3.1.3. Selection Sort

Similar results occur with Selection Sort, as indicated by Table 5.

#### 3.1.4. Merge Sort

Table 6, as with the other algorithms, shows no significant differences in execution time. However, since Merge Sort is the first algorithm in this selection to require additional memory, Table 7 becomes the main focus of this comparison. It shows the free memory available before execution and the algorithm’s memory usage. Considering that the Arduino Uno has only 2048 bytes of SRAM, and the n=1000-sized arrays proved to be problematic even for in-place sorting algorithms, Merge Sort could only be executed up to n=250. Arduino Leonardo, with its 512 B SRAM increase (total 2560 B), was able to handle n=500. The Mega 2560 had no issues and successfully executed all input sizes, with 3933 B of SRAM still free at n=1000.

#### 3.1.5. Quick Sort

Table 8 shows the execution times for Quick Sort across all three Arduino boards. However, as with Merge Sort, the results in Table 9, which presents memory usage, are of greater significance. For randomly ordered data, all boards could execute Quick Sort up to *n* = 750; at *n* = 1000, the Arduino Uno ran out of SRAM, whereas the Leonardo and Mega 2560 still completed successfully. In contrast, the ascending order input proved impossible to execute on both the Arduino Uno and Arduino Leonardo, while the Arduino Mega 2560 managed to process arrays up to n=250, utilizing approximately 83.5% of its available memory. It yielded similar results for the reverse order input, handling up to n=250 but failing at n=500, despite using only 41.9% of its memory at n=250. Both the Uno and Leonardo successfully completed sorting only for n=100.

#### 3.1.6. Heap Sort

From Table 10, it is clear that, as with the other algorithms, Heap Sort’s execution time remained nearly identical across all boards. In terms of memory, Table 11 shows that none of the boards encountered difficulties executing the algorithm. The fixed amount of memory required by Heap Sort allowed all boards, including the Arduino Uno, to successfully execute all tested input sizes supported by the board’s SRAM constraints.

### 3.2. Aggregated Performance Analysis

As expected from the asymptotic behavior summarized in Table 1, the $O(n^{2})$ algorithms quickly become impractical for larger array sizes on 8-bit AVR microcontrollers, whereas the O(nlogn) algorithms remain usable up to n=1000, subject to SRAM limitations. 

#### 3.2.1. Bubble Sort

As shown in Table 12, Bubble Sort’s execution time grew quadratically with the array size, as expected from the theoretical $O(n^{2})$ complexity. When provided with an already sorted array, it terminated almost immediately. The reverse-order runs were, on average, about 57% more time-consuming than the random-order runs. The number of writes also followed a quadratic trend, as presented in Figure 1,with the reverse order requiring roughly twice as many writes as the random case. Owing to its in-place nature, Bubble Sort required no additional memory and executed correctly on all boards for every input size that fit into SRAM. The runtime is shown in Figure 2. The error bars in the Figures represent ±1 standard deviation as described in Section 2.3. 

#### 3.2.2. Insertion Sort

Both the execution time and the number of writes (Table 13) grew quadratically, as expected. Provided with a sorted array, it performed n−1 writes and finished almost immediately. For reverse-order inputs, the execution time and the number of writes were both about twice as high as for the random case (≈100% increase), as shown in Figure 3 and Figure 4. Insertion Sort used no additional memory; therefore, it executed on all three boards. 

#### 3.2.3. Selection Sort

Selection Sort’s execution time increased quadratically, in line with the expected behavior, as detailed in Table 14. There were no observable differences in execution time or number of writes between random, ascending, and reverse-ordered arrays (Figure 5 and Figure 6), since the algorithm performs the same sequence of comparisons regardless of the initial order. Requiring no additional memory, it ran perfectly on all boards.

#### 3.2.4. Merge Sort

As expected from the theoretical complexity O(nlogn), the execution time of the Merge Sort increased approximately linearly over the tested range of array sizes, as shown in Table 15 and Figure 7. It was unaffected by the initial array order and performed with the same efficiency. However, due to its significant memory requirements (Figure 8), not all boards were able to execute it successfully. Arduino Mega 2560, the most capable board of this selection, completed all the tested array sizes. Arduino Leonardo failed at n=750, and Arduino Uno was unable to run arrays of size n=500 or greater. The number of writes followed a similar trend, as illustrated in Figure 9.

#### 3.2.5. Quick Sort

For the random order array, Quick Sort’s theoretical O(nlogn) complexity was confirmed, as demonstrated in Table 16 and Figure 10. Both the ascending and descending order datasets, however, did not provide enough data to observe a clear trend. Due to the pivot selection strategy, performance degraded rapidly for these ordered cases, which is reflected in the dramatic increases in writes (Figure 11) and memory usage (Figure 12). Even the Arduino Mega 2560 failed to execute arrays beyond n=500. In contrast, the Arduino Uno and Leonardo were unable to complete the ascending order tests at all, and for the reverse order, they managed to process only n=100. 

#### 3.2.6. Heap Sort

Heap Sort exhibited a growth in execution time consistent with its theoretical O(nlogn) complexity, as seen in Table 17. It showed similar performance across random, ascending, and reverse order inputs. Executed on an ascending ordered dataset, its runtime was 107% of the random order’s time (Figure 13), with the number of writes at 108% of the random case (Figure 14). Reverse order was the quickest, at 93% of the random order’s time and 90% of the writes. This unexpected phenomenon occurred because, in the reverse-ordered dataset, all the data already largely satisfied the heap property, whereas in the ascending order dataset, many more elements had to be repositioned to build a heap. Memory usage was consistent at 12 bytes for both the Arduino Uno and Leonardo, and 13 bytes for the Arduino Mega 2560. Table 17 highlights a minor inconsistency in memory usage at n=1000, which occurred due to the Arduino Uno being unable to execute algorithms on arrays of that size, and thus could not contribute to the averaged result. Apart from this, Heap Sort executed successfully on all tested boards.

### 3.3. Execution Time Comparison

#### 3.3.1. Random Order

When executed on a randomized array, each algorithm performed as expected (Figure 15). With Merge Sort, Quick Sort, and Heap Sort working efficiently, with the time scaling according to their O(nlogn) complexity. Quick Sort was the fastest, at n=1000, it outperformed Heap Sort by 41.3% and Merge Sort by 43.4%. The slower, quadratic algorithms, Insertion Sort and Selection Sort, showed similar performance, with Insertion Sort being 4% faster. However, even the slowest O(nlogn) algorithm, Merge Sort, was 90.9% more efficient than Insertion Sort. Bubble Sort was by far the least efficient overall, taking 93% longer than Selection Sort and 3508.9% longer than the fastest Quick Sort.

#### 3.3.2. Ascending Order

When provided with an already sorted array (Figure 16), both Bubble Sort and Insertion Sort completed almost instantly, with Bubble Sort executing 66.2% faster. Heap Sort and Merge Sort followed, showing similar performance—Merge Sort being 9.7% faster than Heap Sort. However, when compared to Bubble Sort, Merge Sort required 5047.3% more time. Selection Sort was the slowest of all successfully executed algorithms, taking 64550% longer than Bubble Sort and 929% longer than Heap Sort. Quick Sort failed to execute on any of the boards, preventing its inclusion in this comparison.

#### 3.3.3. Reverse Order

The descending order (Figure 17) proved not to be a problem for both Heap Sort and Merge Sort. They were quick, with Heap Sort being faster than Merge Sort by 5.7%. Selection Sort took 1009.9% more time than Merge Sort, 2040.4% more than Insertion Sort, and 3278.6% more than Bubble Sort, making it by far the worst one in this case.

### 3.4. Number of Writes Comparison

#### 3.4.1. Random Order

In terms of the number of writes, Heap Sort, Quick Sort, Merge Sort, and Selection Sort performed the best when provided with a randomly ordered dataset (Figure 18). Selection Sort had the lowest write count, requiring 80% fewer writes than Merge Sort, 83.5% fewer than Quick Sort, and 89% fewer than Heap Sort. Insertion Sort was significantly less efficient in this regard, producing 12,353% more writes. The least efficient was Bubble Sort, with 24,671.9% more write operations than Selection Sort.

#### 3.4.2. Ascending Order

When provided with a sorted array (Figure 19), Bubble Sort performed no write operations. Insertion Sort was second best, producing 50% fewer writes than Selection Sort, 90% fewer than Merge Sort, and 94.9% fewer than Heap Sort, making Heap Sort the least efficient in this scenario. Since Quick Sort did not run successfully for array sizes larger than n=250, it cannot be directly compared. However, at n=250, it performed 223.2% more write operations than Bubble Sort did at n=1000. 

#### 3.4.3. Reverse Order

The algorithm that performed the fewest write operations when provided with a reversed array was Selection Sort (Figure 20). It was better in this regard than Heap Sort by 88%, Insertion Sort by 99.6%, and Bubble Sort by 99.8%.

### 3.5. Memory Usage Comparison

#### 3.5.1. Random Order

When it comes to memory usage (Figure 21), the most efficient algorithms were those that required little to no additional memory. Bubble Sort, Insertion Sort, and Selection Sort operated entirely in-place, using no extra memory. Another highly performing algorithm was Heap Sort, which utilized only a constant amount of memory. Of the two algorithms that require a lot of additional memory, Quick Sort required 84.7% less, and, unlike Merge Sort, it could still be executed up to *n* = 1000 on the Leonardo and Mega 2560.

#### 3.5.2. Ascending Order

When provided with an already sorted array (Figure 22), all algorithms apart from Quick Sort used exactly the same amount that they did when processing the random order. Quick Sort ran out of memory instantly, only being able to provide data up to n=250. At n=250, it used 1091.8% more memory than Merge Sort at that same *n*.

#### 3.5.3. Reverse Order

Just as with the ascending order, the reverse order did not change the memory usage for any algorithms except Quick Sort (Figure 23). In the reverse-order case, Quick Sort used less memory than for ascending inputs, but it still consumed 498.3% more memory than Merge Sort at n=250. It was still unable to produce any data for larger arrays.

### 3.6. Summary and Interpretation of the Results

#### 3.6.1. Performance Analysis

All $O(n^{2})$ complexity algorithms executed successfully for the input sizes that fit into the available SRAM on each board, but they were considerably slower than the O(nlogn) algorithms. While the latter demonstrated higher efficiency, they also required additional memory, which posed challenges in a memory-constrained environment.

Among the $O(n^{2})$ complexity algorithms, Bubble Sort was consistently the slowest and performed the highest amount of write operations. Although it performed exceptionally well on pre-sorted arrays, this scenario can be overlooked since it almost never occurs in practice. Insertion Sort and Selection Sort were faster, with similar execution times on randomly ordered data. Selection Sort was consistent with both time and number of writes across all order types, whereas Insertion Sort was heavily influenced by the order. Since none of the mentioned algorithms requires memory, they are suitable for microcontrollers with constrained memory.

The O(nlogn) algorithms—Merge Sort, Quick Sort, and Heap Sort—were substantially faster and required fewer write operations overall. However, their increased memory demands significantly limited their usability. Merge Sort required the most memory and thus failed to execute at larger input sizes on both the Arduino Uno and Leonardo. It failed consistently regardless of the initial array order. Quick Sort performed best on randomly ordered arrays, but at *n*=1000 it could only be executed on the Leonardo and Mega 2560, while the Uno ran out of memory. It, however, became highly inefficient on ordered datasets, as poor pivot selection led to deep recursion and excessive memory usage. Such situations are uncommon but may still lead to program instability on rare occasions. In contrast, Heap Sort achieved consistent results across all order types and boards, maintaining stable memory usage and predictable performance, making it the most balanced choice among the O(nlogn) group for resource-constrained devices. 

#### 3.6.2. Practical Implications

For low-memory microcontroller applications, both Selection Sort and Insertion Sort offer reliable and memory-efficient solutions, albeit at the expense of execution speed. Bubble Sort performs similarly but is even less efficient, making it unsuitable for most practical applications. Heap Sort offers an optimal balance between performance, memory usage, and predictability, making it the best choice among these algorithms. Merge Sort is impractical for memory-constrained environments and should be avoided. Lastly, while Quick Sort performs well in theory, it should be avoided due to its significant memory consumption when partially ordered input arrays are introduced.

## 4. Discussion

The obtained results are consistent with previous work that emphasizes the memory–speed trade-off in sorting on embedded systems [3,4,5]. They demonstrate that, under strict SRAM limits typical for 8-bit AVR Arduino-class sensors and edge prototypes, lower asymptotic time-complexity algorithms can fall behind simpler $O(n^{2})$ methods once their additional memory requirements are taken into account. This challenge of balancing computation and memory access patterns was also highlighted in Lawrence’s adaptive flash sorting research [5], and our findings confirm that it extends to in-RAM sorting on low-end microcontrollers.

Among the tested algorithms, Heap Sort demonstrated the most favorable compromise for sensor and edge devices. It maintained the preferable O(nlogn) behavior while using a fixed and predictable amount of memory, and it executed reliably across all tested input sizes and orderings. This aligns with the principles behind adaptive and external sorting techniques, which prioritize bounded memory usage and regular access patterns over marginal improvements in runtime [3].

### Limitations and Generality of the Results

The results reported in this study should be interpreted in light of several important limitations. First, all experiments were performed on low-end 8-bit AVR microcontrollers running at 16 MHz, namely the Arduino Uno, Arduino Leonardo, and Arduino Mega 2560. These platforms are representative of widely used educational and prototyping boards, but they do not cover the much broader spectrum of contemporary IoT and edge devices, many of which rely on 32-bit Cortex-M or DSP-class microcontrollers with substantially larger SRAM and flash memories and, in some cases, hardware floating-point units.

Second, only signed integer datasets were considered, chosen to reflect typical usage on Arduino-class boards and to avoid complications related to software-emulated floating-point arithmetic. The behavior of the examined algorithms on floating-point data, or on architectures with efficient hardware support for floating-point operations, may differ significantly from the integer-only results reported here.

Third, the experiments focus on in-memory sorting of contiguous arrays and do not include genuinely external sorting scenarios, where data reside primarily in flash memory or on external storage. For such settings, specialized flash-oriented sorting algorithms have been shown to provide substantial benefits over straightforward in-RAM methods [3,4,5,6,7,8]. Our measurements therefore complement, rather than replace, those external-memory studies by characterizing the in-SRAM behavior of classical sorting algorithms on very memory-constrained devices.

Finally, the experimental design does not explicitly model energy consumption at the hardware level. We instead use the number of array-element writes as a coarse proxy for write-related energy and wear, motivated by previous work on flash-based embedded sorting. While this metric is informative for comparing algorithms qualitatively, a full energy characterization would require direct current measurements using external hardware instrumentation (e.g., oscilloscopes or power analyzers) and a detailed power model of the microcontroller and memory subsystems, which are beyond the scope of this study.

Overall, our results suggest that the practical efficiency of sorting algorithms in memory-constrained sensing environments is more strongly governed by space complexity and memory-access patterns than by theoretical time complexity alone. For typical 8-bit AVR Arduino-class platforms that must locally manage queues or reorder sensor readings before transmission, Heap Sort offers a robust default choice, combining predictable memory usage with consistently good performance. Future research could focus on hybrid approaches that combine Heap Sort with lightweight $O(n^{2})$ methods for very small datasets, algorithm variants explicitly optimized for energy consumption and flash wear, and extending the evaluation to 32-bit IoT platforms and real sensor workloads, including streaming and sliding-window scenarios.

## 5. Conclusions

This paper presents an empirical comparison of six classic sorting algorithms—Bubble Sort, Insertion Sort, Selection Sort, Merge Sort, QuickSort, and Heap Sort—on three 8-bit AVR Arduino-class microcontrollers, which are representative of resource-constrained sensing and IoT prototyping nodes. For each algorithm, we measured execution time, array-element write operations, and SRAM usage across a range of array sizes and input orderings. The results quantify how algorithmic choices interact with the tight memory budgets and limited processing power typical for such low-end platforms.

The experiments confirmed that the asymptotically faster O(nlogn) algorithms are not always the most practical option in this environment. Merge Sort and Quick Sort, while fast for small and randomly ordered datasets, quickly became impractical due to their elevated memory requirements and, in the case of Quick Sort, the risk of deep recursion on partially ordered inputs. In contrast, the simpler $O(n^{2})$ algorithms—in particular, Insertion Sort and Selection Sort—remained robust, memory-efficient choices for small datasets, despite their higher runtime. Among all evaluated methods, Heap Sort offered the most favorable compromise, combining predictable and bounded memory usage with consistently good performance across all boards, input sizes, and orderings.

These findings can support practitioners in selecting sorting algorithms for embedded sensing applications implemented on 8-bit AVR Arduino-class platforms and other similar low-end microcontrollers used in IoT nodes, where on-node pre-processing must be balanced against severe memory and energy limitations. Future work may extend this study to 32-bit IoT platforms, richer sensor workloads (including streaming and sliding-window scenarios), and algorithm variants explicitly optimized for energy consumption and non-volatile memory wear. Specifically, for N=1000 random elements, Heap Sort (approx. 92ms) offered a 21x speedup over Bubble Sort (approx. 1958ms) while consuming negligible SRAM, proving to be the optimal choice for the examined 8-bit platforms.

## Figures and Tables

**Figure 1 sensors-26-00214-f001:**
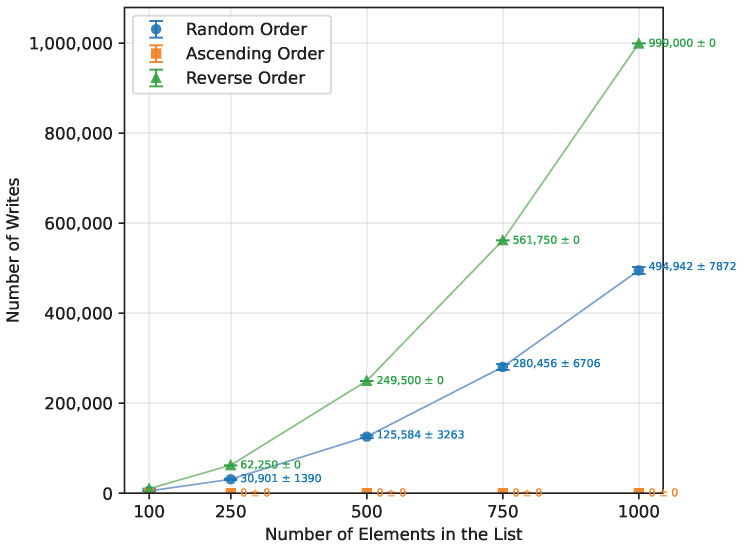
Bubble Sort number of writes vs. array size (averaged across boards).

**Figure 2 sensors-26-00214-f002:**
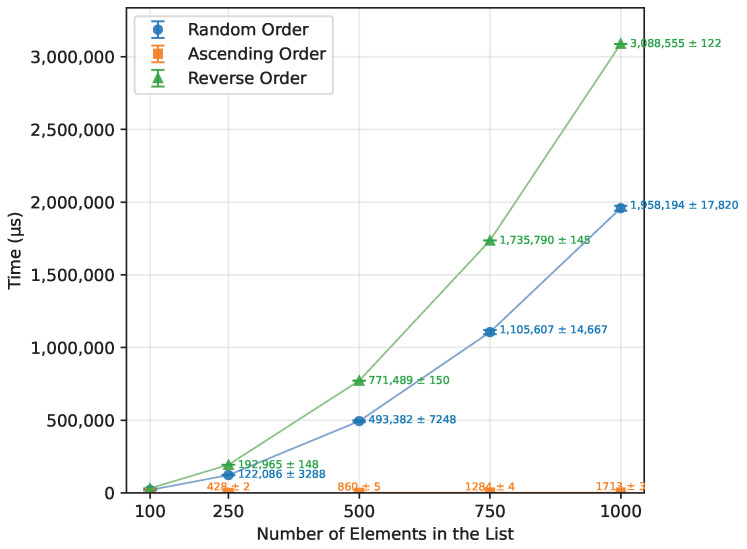
Bubble Sort runtime vs. array size (averaged across boards).

**Figure 3 sensors-26-00214-f003:**
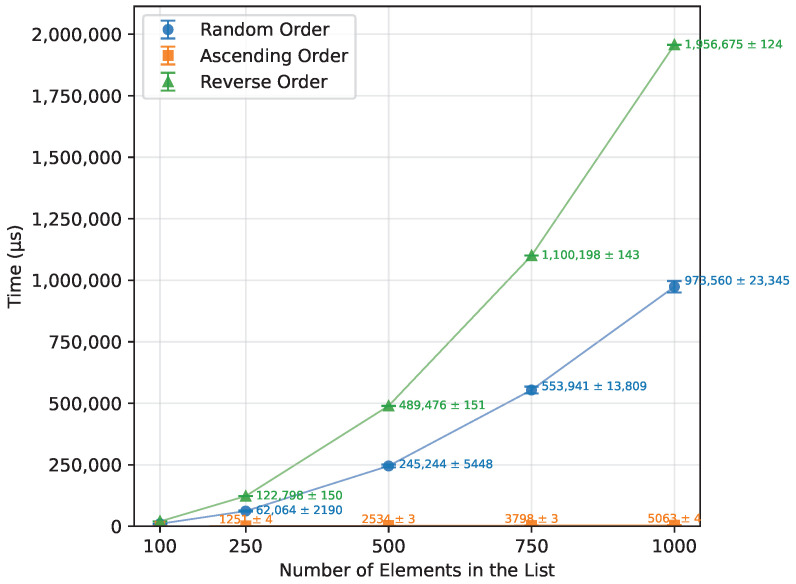
Insertion Sort runtime vs. array size (averaged across boards).

**Figure 4 sensors-26-00214-f004:**
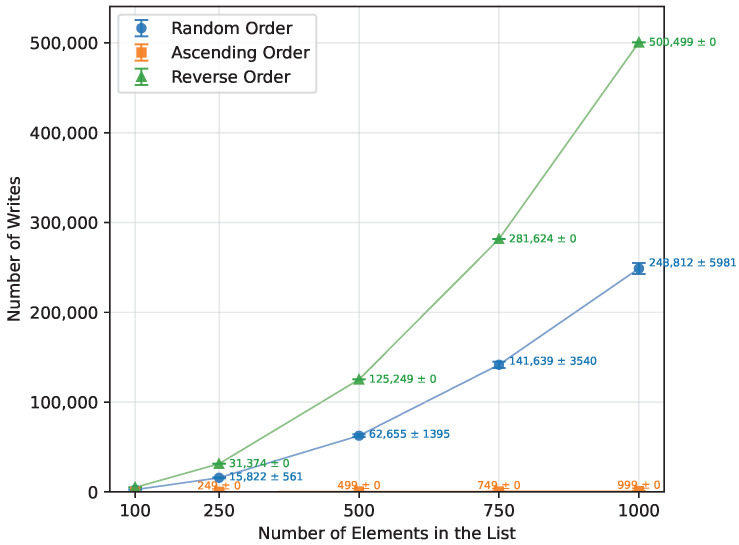
Insertion Sort number of writes vs. array size (averaged across boards).

**Figure 5 sensors-26-00214-f005:**
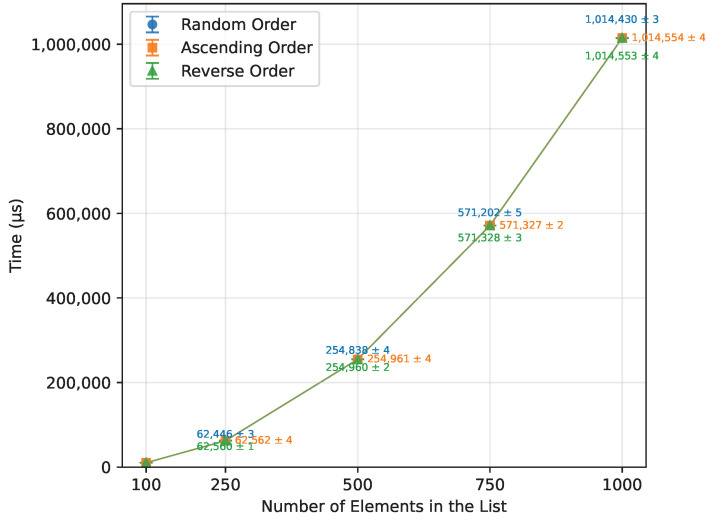
Selection Sort runtime vs array size (averaged across boards).

**Figure 6 sensors-26-00214-f006:**
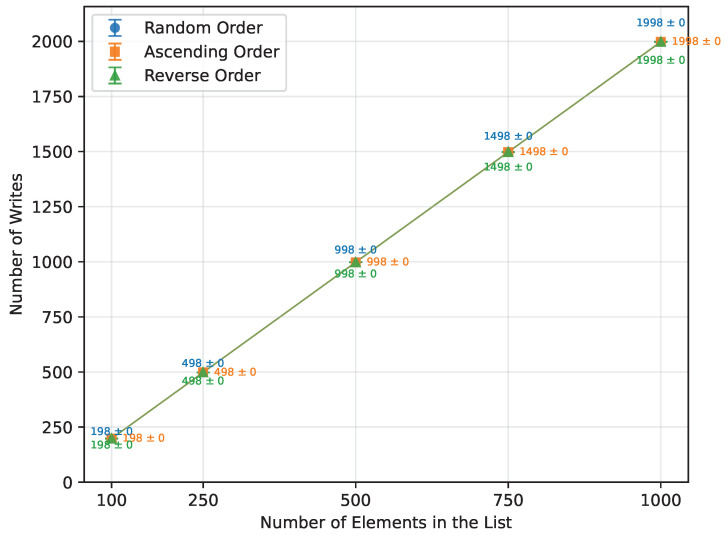
Selection Sort number of writes vs. array size (averaged across boards).

**Figure 7 sensors-26-00214-f007:**
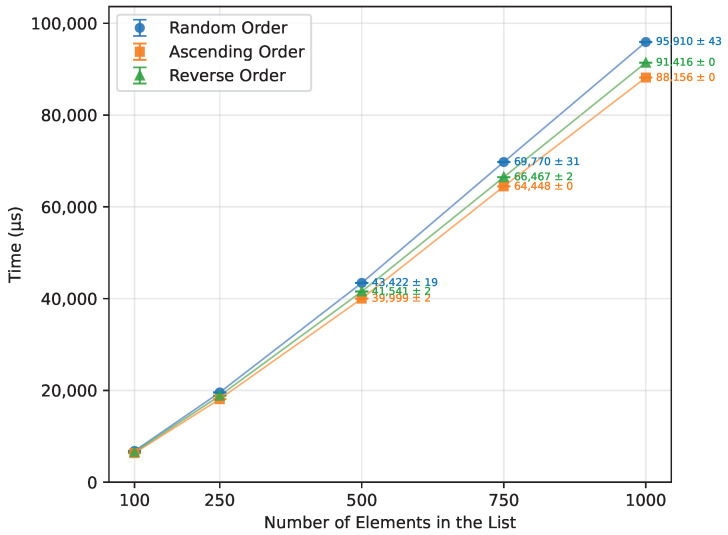
Merge Sort runtime vs. array size (averaged across boards).

**Figure 8 sensors-26-00214-f008:**
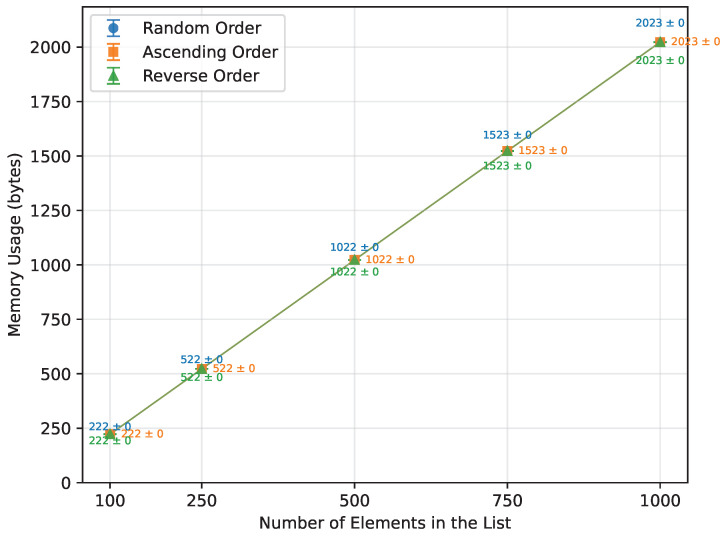
Merge Sort memory usage vs. array size (averaged across boards).

**Figure 9 sensors-26-00214-f009:**
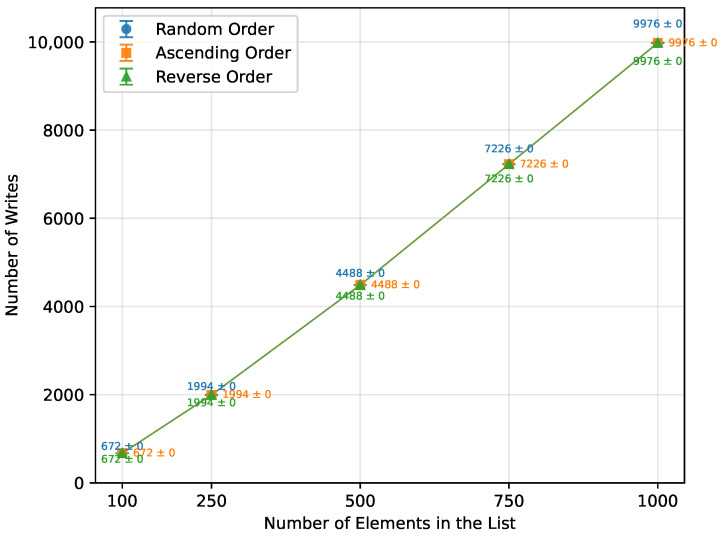
Merge Sort number of writes vs. array size (averaged across boards).

**Figure 10 sensors-26-00214-f010:**
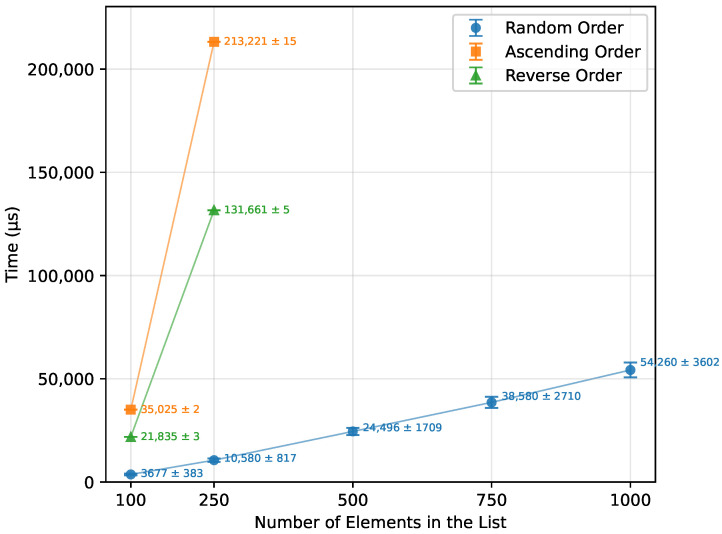
Quick Sort runtime vs. array size (averaged across boards).

**Figure 11 sensors-26-00214-f011:**
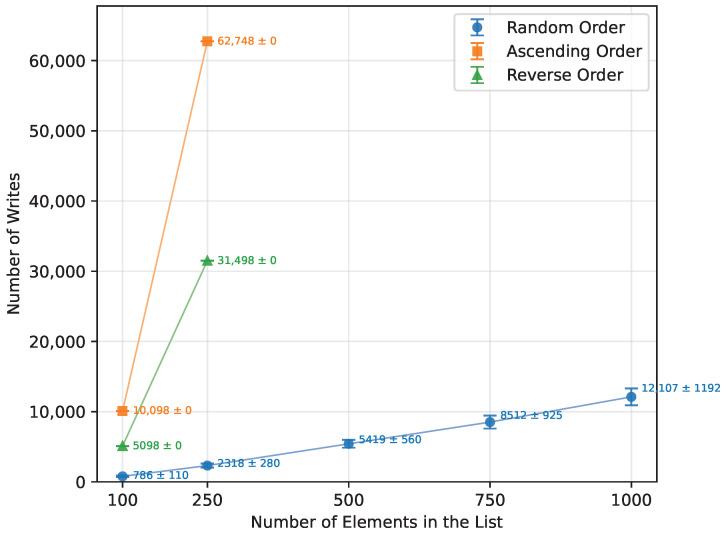
Quick Sort number of writes vs. array size (averaged across boards).

**Figure 12 sensors-26-00214-f012:**
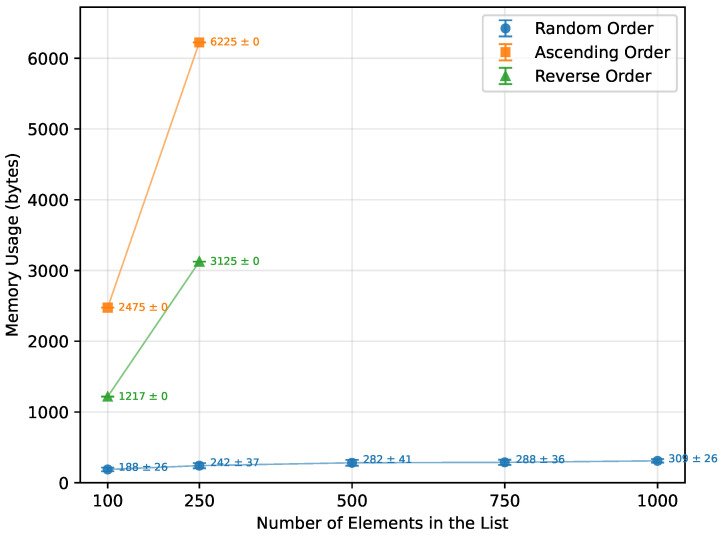
Quick Sort memory usage vs. array size (averaged across boards).

**Figure 13 sensors-26-00214-f013:**
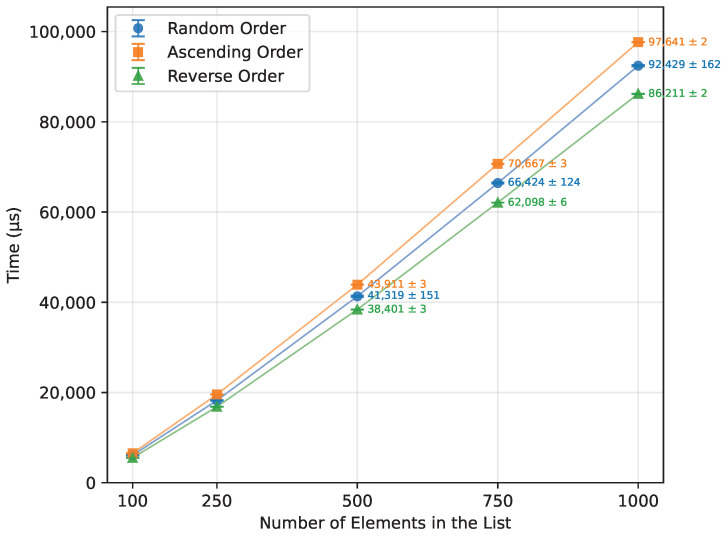
Heap Sort runtime vs. array size (averaged across boards).

**Figure 14 sensors-26-00214-f014:**
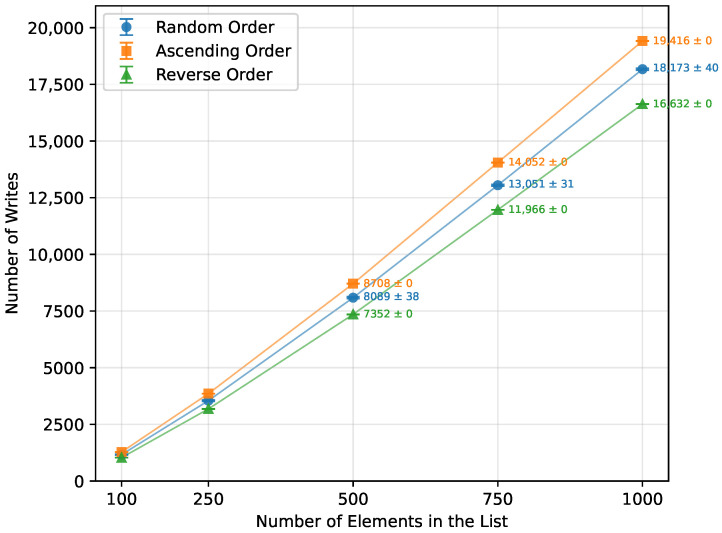
Heap Sort number of writes vs. array size (averaged across boards).

**Figure 15 sensors-26-00214-f015:**
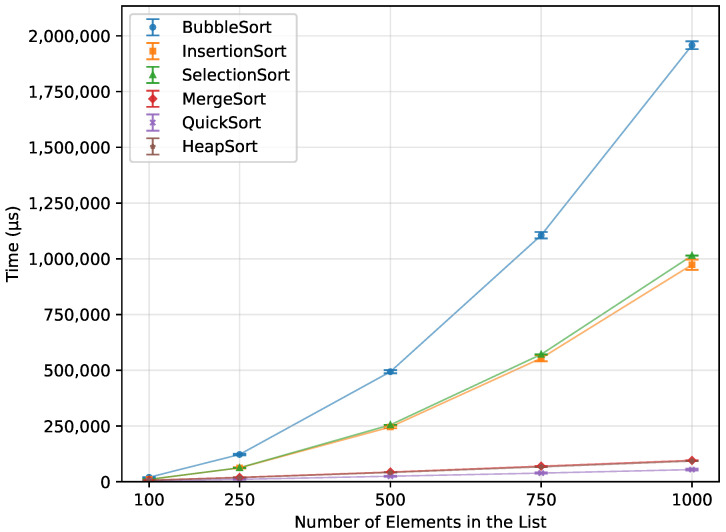
Execution time of all algorithms vs. array size for random-order inputs (averaged across boards).

**Figure 16 sensors-26-00214-f016:**
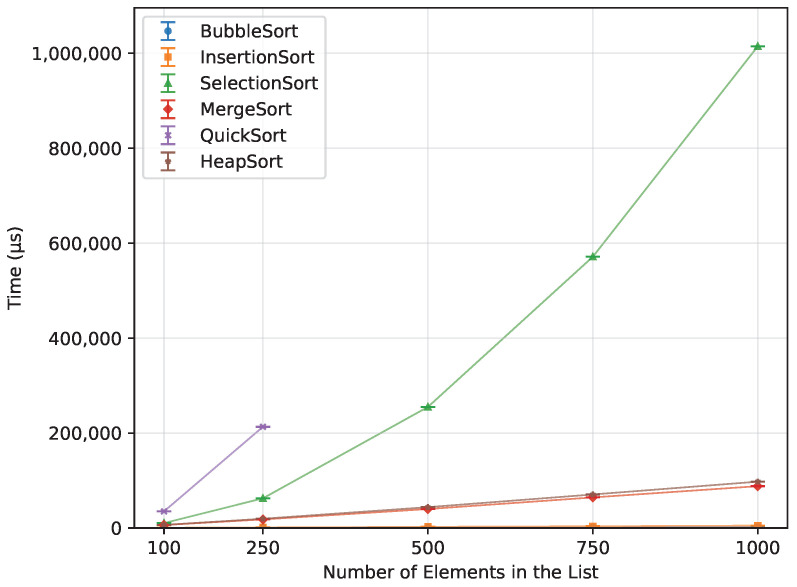
Execution time of all algorithms vs. array size for ascending-order inputs (averaged across boards).

**Figure 17 sensors-26-00214-f017:**
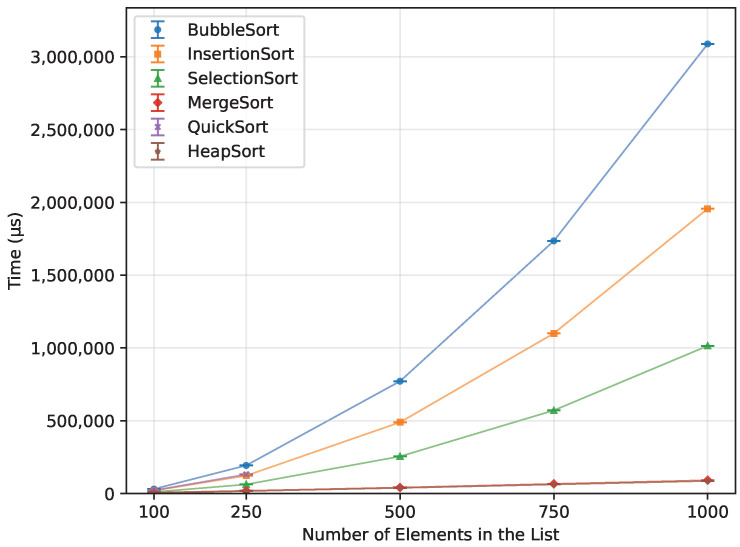
Execution time of all algorithms vs. array size for reverse-order inputs (averaged across boards).

**Figure 18 sensors-26-00214-f018:**
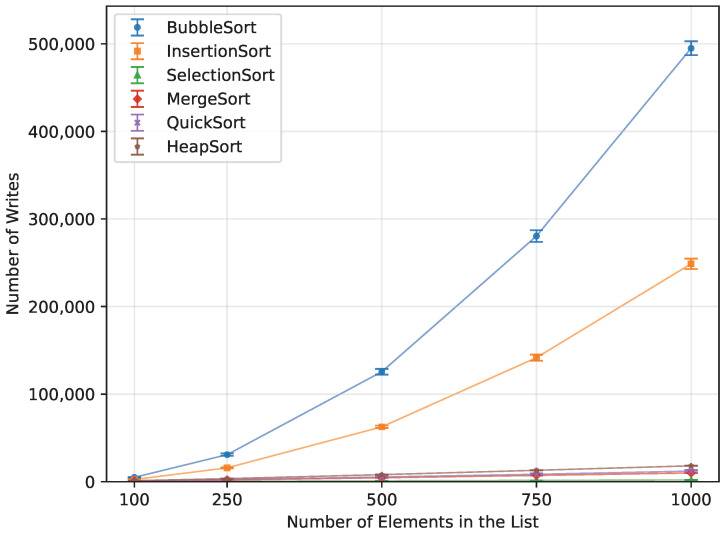
Number of array-element writes for all algorithms vs. array size for random-order inputs (averaged across boards).

**Figure 19 sensors-26-00214-f019:**
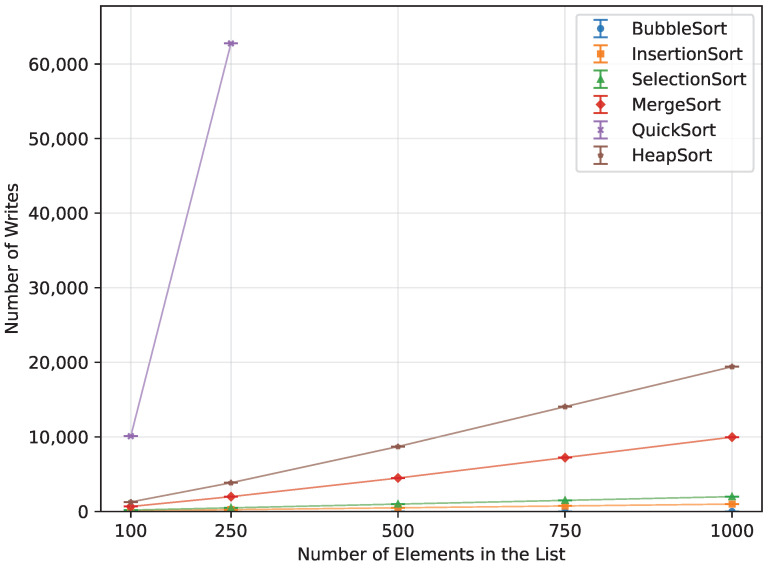
Number of array-element writes for all algorithms vs. array size for ascending-order inputs (averaged across boards).

**Figure 20 sensors-26-00214-f020:**
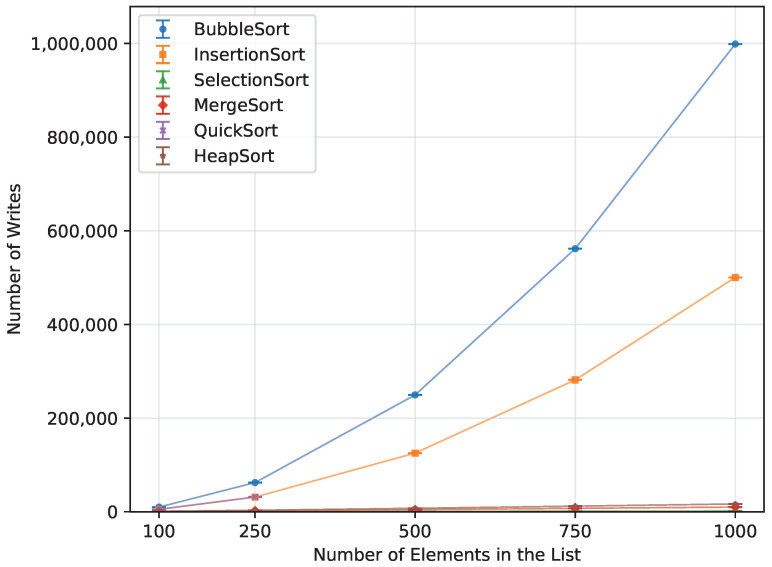
Number of array-element writes for all algorithms vs. array size for reverse-order inputs (averaged across boards).

**Figure 21 sensors-26-00214-f021:**
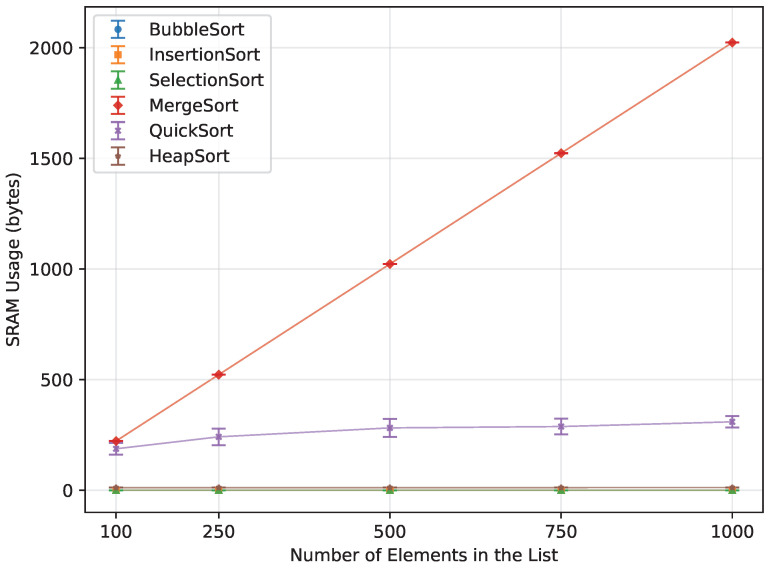
SRAM usage of all algorithms vs. array size for random-order inputs (averaged across boards).

**Figure 22 sensors-26-00214-f022:**
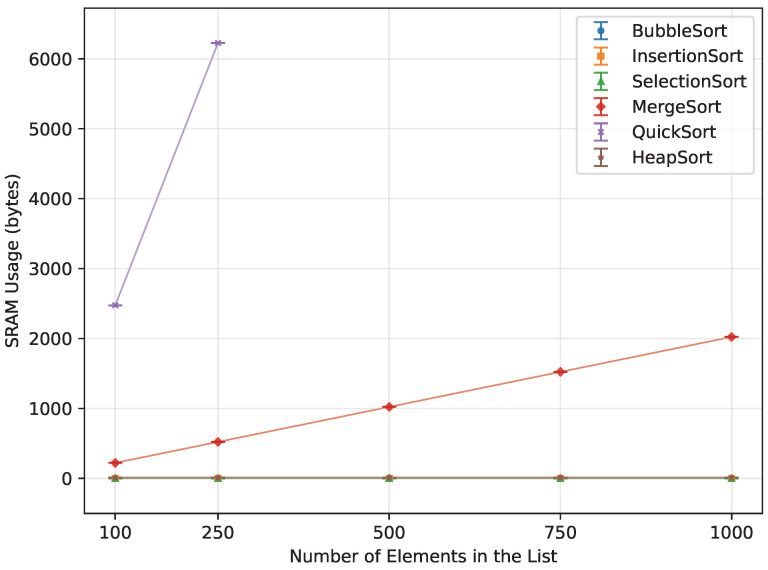
SRAM usage of all algorithms vs. array size for ascending-order inputs (averaged across boards).

**Figure 23 sensors-26-00214-f023:**
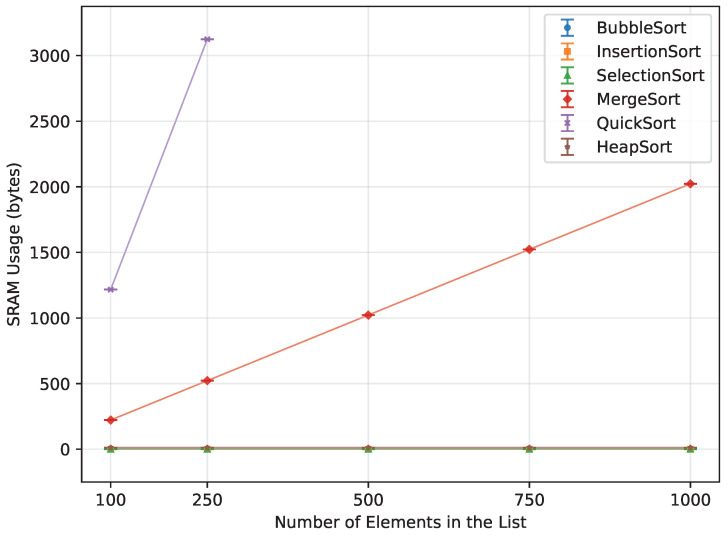
SRAM usage of all algorithms vs. array size for reverse-order inputs (averaged across boards.

**Table 1 sensors-26-00214-t001:** Summary of theoretical properties of the sorting algorithms considered in this study. Time complexity is given in terms of input size *n*; extra memory refers to auxiliary storage beyond the input array. The complexity and stability characteristics follow standard textbook results for comparison-based sorting algorithms [21,22,23].

Algorithm	Time Complexity (Best/Average/Worst)	Extra Memory	Stability
Bubble Sort	O(n) / $O(n^{2})$ / $O(n^{2})$	O(1) (in-place)	stable
Insertion Sort	O(n) / $O(n^{2})$ / $O(n^{2})$	O(1) (in-place)	stable
Selection Sort	$O(n^{2})$ / $O(n^{2})$ / $O(n^{2})$	O(1) (in-place)	not stable
Merge Sort	O(nlogn) / O(nlogn) / O(nlogn)	O(n) (auxiliary array)	stable
Quick Sort	O(nlogn) / O(nlogn) / $O(n^{2})$	O(logn) (recursion stack)	not stable
Heap Sort	O(nlogn) / O(nlogn) / O(nlogn)	O(1) (in-place)	not stable

**Table 2 sensors-26-00214-t002:** Measured versus theoretical SRAM usage for varying recursion depths and local array sizes.

Recursion Depth	Array Size [B]	Free Memory [B]	Used Memory [B]	Theoretical [B]	Offset [B]
1	4	7913	40	4	36
1	16	7901	52	16	36
1	64	7853	100	64	36
1	128	7789	164	128	36
10	4	7796	157	40	117
10	16	7676	277	160	117
10	64	7196	757	640	117
10	128	6556	1397	1280	117
25	4	7601	352	100	252
25	16	7301	652	400	252
25	64	6101	1852	1600	252
25	128	4501	3452	3200	252
50	4	7276	677	200	477
50	16	6676	1277	800	477
50	64	4276	3677	3200	477
50	128	1076	6877	6400	477

**Table 3 sensors-26-00214-t003:** Bubble Sort execution time comparison across Arduino Uno, Leonardo, and Mega2560.

Order Type	N	Uno [μs]	Leonardo [μs]	Mega 2560 [μs]
Random Order	100	19,869.00	19,190.50	20,052.50
	250	122,663.50	119,963.50	123,630.00
	500	493,250.50	492,573.00	494,323.50
	750	1,111,438.00	1,103,227.50	1,102,154.50
	1000	–	1,964,432.00	1,951,955.50
Ascending Order	100	174.67	173.33	172.00
	250	432.00	425.33	426.67
	500	857.33	860.00	861.33
	750	1286.67	1282.67	1284.00
	1000	–	1713.33	1712.00
Reverse Order	100	30,989.33	30,926.67	31,012.00
	250	192,697.33	193,449.33	192,748.00
	500	770,197.33	773,946.67	770,322.67
	750	1,732,793.33	1,741,532.00	1,733,044.00
	1000	–	3,096,206.67	3,080,904.00

**Table 4 sensors-26-00214-t004:** Insertion Sort execution time comparison across Arduino Uno, Leonardo, and Mega2560.

Order Type	N	Uno [μs]	Leonardo [μs]	Mega 2560 [μs]
Random Order	100	9648.00	10,249.50	10,639.50
	250	64,666.00	61,887.00	59,640.00
	500	246,226.50	246,895.00	242,610.50
	750	555,030.00	548,295.50	558,498.50
	1000	-	959,124.00	987,995.50
Ascending Order	100	500.00	498.67	497.33
	250	1246.67	1254.67	1250.67
	500	2534.67	2530.67	2536.00
	750	3793.33	3796.00	3805.33
	1000	-	5056.00	5070.67
Reverse Order	100	19,860.00	19,822.67	19,874.67
	250	122,658.67	123,040.00	122,694.67
	500	488,682.67	490,970.67	488,773.33
	750	1,098,326.67	1,103,774.67	1,098,492.00
	1000	-	1,961,472.00	1,951,877.33

**Table 5 sensors-26-00214-t005:** Selection Sort execution time comparison across Arduino Uno, Leonardo, and Mega2560.

Order Type	N	Uno [μs]	Leonardo [μs]	Mega 2560 [μs]
Random Order	100	10,378.50	10,371.50	10,386.00
	250	62,327.00	62,677.00	62,334.50
	500	254,380.00	255,722.00	254,411.00
	750	570,189.00	573,157.00	570,259.00
	1000	–	1,016,991.50	1,011,868.00
Ascending Order	100	10,388.00	10,365.33	10,400.00
	250	62,493.33	62,677.33	62,516.00
	500	254,538.67	255,744.00	254,600.00
	750	570,346.67	573,186.67	570,448.00
	1000	–	1,017,032.00	1,012,076.00
Reverse Order	100	10,388.00	10,369.33	10,400.00
	250	62,492.00	62,676.00	62,513.33
	500	254,540.00	255,741.33	254,598.67
	750	570,346.67	573,190.67	570,446.67
	1000	–	1,017,033.33	1,012,073.33

**Table 6 sensors-26-00214-t006:** Merge Sort execution time comparison across Arduino Uno, Leonardo, and Mega2560.

Order Type	N	Uno [μs]	Leonardo [μs]	Mega 2560 [μs]
Random Order	100	6756.00	6753.00	6900.00
	250	19,377.50	19,477.50	19,801.50
	500	–	43,057.00	43,786.00
	750	–	–	69,770.50
	1000	–	–	95,909.50
Ascending Order	100	6298.67	6281.33	6446.67
	250	17,982.67	17,949.33	18,422.67
	500	–	39,538.67	40,460.00
	750	–	–	64,448.00
	1000	–	–	88,156.00
Reverse Order	100	6512.00	6501.33	6660.00
	250	18,710.67	18,676.00	19,152.00
	500	–	41,089.33	41,992.00
	750	–	–	66,466.67
	1000	–	–	91,416.00

**Table 7 sensors-26-00214-t007:** Merge Sort starting free memory and memory usage comparison across Arduino Uno, Leonardo, and Mega2560.

Order Type	N	Uno		Leonardo		Mega 2560	
		Free [B]	Used [B]	Free [B]	Used [B]	Free [B]	Used [B]
Random Order	100	1614	222	2161	222	7756	223
	250	1314	522	1861	522	7456	523
	500	-	-	1361	1022	6956	1023
	750	-	-	-	-	6456	1523
	1000	-	-	-	-	5956	2023
Ascending Order	100	1614	222	2161	222	7756	223
	250	1314	522	1861	522	7456	523
	500	-	-	1361	1022	6956	1023
	750	-	-	-	-	6456	1523
	1000	-	-	-	-	5956	2023
Reverse Order	100	1614	222	2161	222	7756	223
	250	1314	522	1861	522	7456	523
	500	-	-	1361	1022	6956	1023
	750	-	-	-	-	6456	1523
	1000	-	-	-	-	5956	2023

**Table 8 sensors-26-00214-t008:** Quick Sort execution time comparison across Arduino Uno, Leonardo, and Mega2560.

Order Type	N	Uno [μs]	Leonardo [μs]	Mega 2560 [μs]
Random Order	100	3759.00	3684.00	3588.00
	250	10,226.00	10,521.00	10,994.50
	500	23,964.50	24,560.00	24,962.50
	750	36,940.00	40,565.00	38,233.50
	1000	–	53,922.50	54,598.50
Ascending Order	100	–	–	35,025.33
	250	–	–	213,221.33
	500	–	–	–
	750	–	–	–
	1000	–	–	–
Reverse Order	100	21,837.33	21,798.67	21,868.00
	250	–	–	131,661.33
	500	–	–	–
	750	–	–	–
	1000	–	–	–

**Table 9 sensors-26-00214-t009:** Quick Sort starting free memory and memory usage comparison across Arduino Uno, Leonardo, and Mega2560.

Order Type	N	Uno		Leonardo		Mega 2560	
		Free [B]	Used [B]	Free [B]	Used [B]	Free [B]	Used [B]
Random Order	100	1612	186	2159	189	7759	188
	250	1312	219	1861	243	7459	263
	500	812	273	1361	279	6957	294
	750	314	252	863	312	6457	300
	1000	-	-	361	303	5957	316
Ascending Order	100	-	-	-	-	7759	2475
	250	-	-	-	-	7459	6225
	500	-	-	-	-	-	-
	750	-	-	-	-	-	-
	1000	-	-	-	-	-	-
Reverse Order	100	1612	1200	2159	1200	7759	1250
	250	-	-	-	-	7459	3125
	500	-	-	-	-	-	-
	750	-	-	-	-	-	-
	1000	-	-	-	-	-	-

**Table 10 sensors-26-00214-t010:** Heap Sort execution time comparison across Arduino Uno, Leonardo, and Mega2560.

Order Type	N	Uno [μs]	Leonardo [μs]	Mega 2560 [μs]
Random Order	100	6094.50	6085.00	6097.50
	250	18,144.50	18,313.00	18,275.50
	500	41,262.00	41,388.50	41,306.50
	750	66,207.50	66,597.50	66,466.00
	1000	-	92,554.50	92,303.50
Ascending Order	100	6530.67	6520.00	6562.67
	250	19,544.00	19,504.00	19,626.67
	500	43,824.00	43,909.33	44,000.00
	750	70,504.00	70,741.33	70,756.00
	1000	-	97,673.33	97,609.33
Reverse Order	100	5518.67	5510.67	5552.00
	250	16,821.33	16,782.67	16,905.33
	500	38,322.67	38,380.00	38,500.00
	750	61,957.33	62,140.00	62,197.33
	1000	-	86,204.00	86,218.67

**Table 11 sensors-26-00214-t011:** Heap Sort starting free memory and memory usage comparison across Arduino Uno, Leonardo, and Mega2560.

Order Type	N	Uno		Leonardo		Mega 2560	
		Free [B]	Used [B]	Free [B]	Used [B]	Free [B]	Used [B]
Random Order	100	1618	12	2165	12	7759	13
	250	1318	12	1865	12	7459	13
	500	818	12	1365	12	6959	13
	750	318	12	865	12	6459	13
	1000	-	-	365	12	5959	13
Ascending Order	100	1618	12	2165	12	7759	13
	250	1318	12	1865	12	7459	13
	500	818	12	1365	12	6959	13
	750	318	12	865	12	6459	13
	1000	-	-	365	12	5959	13
Reverse Order	100	1618	12	2165	12	7759	13
	250	1318	12	1865	12	7459	13
	500	818	12	1365	12	6959	13
	750	318	12	865	12	6459	13
	1000	-	-	365	12	5959	13

**Table 12 sensors-26-00214-t012:** Results for Bubble Sort, averaged across all three boards.

Order Type	N	Time [μs]	Memory Usage [B]	No. Writes
Random Order	100	19,704.00	0.00	4958.17
	250	122,085.67	0.00	30,901.08
	500	493,382.33	0.00	125,583.58
	750	1,105,606.67	0.00	280,455.83
	1000	1,958,193.75	0.00	494,941.63
Ascending Order	100	173.33	0.00	0.00
	250	428.00	0.00	0.00
	500	859.56	0.00	0.00
	750	1284.44	0.00	0.00
	1000	1712.67	0.00	0.00
Reverse Order	100	30,976.00	0.00	9900.00
	250	192,964.89	0.00	62,250.00
	500	771,488.89	0.00	249,500.00
	750	1,735,789.78	0.00	561,750.00
	1000	3,088,555.33	0.00	999,000.00

**Table 13 sensors-26-00214-t013:** Results for Insertion Sort, averaged across all three boards.

Order Type	N	Time [μs]	Memory Usage [B]	No. Writes
Random Order	100	10,179.00	0.00	2566.96
	250	62,064.33	0.00	15,821.83
	500	245,244.00	0.00	62,655.17
	750	553,941.33	0.00	141,639.29
	1000	973,559.75	0.00	248,811.81
Ascending Order	100	498.67	0.00	99.00
	250	1250.67	0.00	249.00
	500	2533.78	0.00	499.00
	750	3798.22	0.00	749.00
	1000	5063.33	0.00	999.00
Reverse Order	100	19,852.44	0.00	5049.00
	250	122,797.78	0.00	31,374.00
	500	489,475.56	0.00	125,249.00
	750	1,100,197.78	0.00	281,624.00
	1000	1,956,674.67	0.00	500,499.00

**Table 14 sensors-26-00214-t014:** Results for Selection Sort, averaged across all three boards.

Order Type	N	Time [μs]	Memory Usage [B]	No. Writes
Random Order	100	10,378.67	0.00	198.00
	250	62,446.17	0.00	498.00
	500	254,837.67	0.00	998.00
	750	571,201.67	0.00	1498.00
	1000	1,014,429.75	0.00	1998.00
Ascending Order	100	10,384.44	0.00	198.00
	250	62,562.22	0.00	498.00
	500	254,960.89	0.00	998.00
	750	571,327.11	0.00	1498.00
	1000	1,014,554.00	0.00	1998.00
Reverse Order	100	10,385.78	0.00	198.00
	250	62,560.44	0.00	498.00
	500	254,960.00	0.00	998.00
	750	571,328.00	0.00	1498.00
	1000	1,014,553.33	0.00	1998.00

**Table 15 sensors-26-00214-t015:** Results for Merge Sort, averaged across all three boards.

Order Type	N	Time [μs]	Memory Usage [B]	No. Writes
Random Order	100	6803.00	222.33	672.00
	250	19,552.17	522.33	1994.00
	500	43,421.50	1022.50	4488.00
	750	69,770.50	1523.00	7226.00
	1000	95,909.50	2023.00	9976.00
Ascending Order	100	6342.22	222.33	672.00
	250	18,118.22	522.33	1994.00
	500	39,999.33	1022.50	4488.00
	750	64,448.00	1523.00	7226.00
	1000	88,156.00	2023.00	9976.00
Reverse Order	100	6557.78	222.33	672.00
	250	18,846.22	522.33	1994.00
	500	41,540.67	1022.50	4488.00
	750	66,466.67	1523.00	7226.00
	1000	91,416.00	2023.00	9976.00

**Table 16 sensors-26-00214-t016:** Results for Quick Sort, averaged across all three boards.

Order Type	N	Time [μs]	Memory Usage [B]	No. Writes
Random Order	100	3677.00	187.50	786.25
	250	10,580.50	241.50	2317.58
	500	24,495.67	281.92	5418.75
	750	38,579.50	288.00	8511.83
	1000	54,260.50	309.31	12,107.00
Ascending Order	100	35,025.33	2475.00	10,098.00
	250	213,221.33	6225.00	62,748.00
	500	–	–	–
	750	–	–	–
	1000	–	–	–
Reverse Order	100	21,834.67	1216.67	5098.00
	250	131,661.33	3125.00	31,498.00
	500	–	–	–
	750	–	–	–
	1000	–	–	–

**Table 17 sensors-26-00214-t017:** Results for Heap Sort, averaged across all three boards.

Order Type	N	Time [μs]	Memory Usage [B]	No. Writes
Random Order	100	6092.33	12.33	1165.33
	250	18,244.33	12.33	3547.08
	500	41,319.00	12.33	8089.33
	750	66,423.67	12.33	13,050.92
	1000	92,429.00	12.50	18,172.75
Ascending Order	100	6537.78	12.33	1280.00
	250	19,558.22	12.33	3856.00
	500	43,911.11	12.33	8708.00
	750	70,667.11	12.33	14,052.00
	1000	97,641.33	12.50	19,416.00
Reverse Order	100	5527.11	12.33	1032.00
	250	16,836.44	12.33	3182.00
	500	38,400.89	12.33	7352.00
	750	62,098.22	12.33	11,966.00
	1000	86,211.33	12.50	16,632.00

## Data Availability

Data (version 1.0) are available in a publicly accessible repository at the GitHub repository sorting-algorithms-on-microcontrollers (accessed on 21 December 2025).

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
