# Peer review of "Empirical Evaluation of Unoptimized Sorting Algorithms on 8-Bit AVR Arduino Microcontrollers"

_sensors, 2025, doi:10.3390/s26010214_

Round 1

Reviewer 1 Report

Comments and Suggestions for Authors

The manuscript presents an experimental evaluation of six sorting algorithms (Bubble, Insertion, Selection, Merge, Quick, Heap) implemented on three Arduino boards (Uno, Leonardo, Mega2560). While the empirical topic has potential relevance for embedded-system educators and practitioners, the current submission does not provide sufficiently rigorous, validated, or reproducible evidence to support the practical recommendations advanced by the authors. On this basis, my recommendation is to reject the manuscript in its present form.
1- Timing and SRAM measurement procedures are unreliable and unvalidated: The manuscript relies on measurement procedures that are not demonstrated to be accurate or repeatable. There is no validation or cross-checking of the timing and memory-measurement methods against higher-resolution instrumentation or well-established benchmarks.
2- Missing compilation and code provenance metadata: The manuscript does not provide essential metadata that would enable reproduction: compiler version, optimization flags, exact source code (or a repository link), build environment, and explicit steps to replicate the experiments. Without these, reproducibility is severely compromised.
3- Insufficient statistical reporting: The results are reported without basic measures of dispersion (standard deviations) or confidence intervals. No explanation of the number of repetitions, outlier handling, or statistical tests is provided. This omission prevents any reliable assessment of effect sizes or the robustness of observed differences.
4- Over-generalized industrial recommendations from limited experiments: The authors make broad recommendations for designers of IoT and embedded systems (e.g., advocating heap sort as a default) on the basis of experiments limited to non-optimized, theoretical implementations on three 8-bit Arduino boards. Such extrapolations to industrial practice are not justified by the presented data.
5- Inadequate timing resolution and validation for short executions: Timing measurements are taken using millis() (1 ms resolution). The manuscript quantifies very short executions using this coarse timer without demonstrating that quantization error does not affect the results. Validation using higher-resolution timers (e.g., micros()) or external instrumentation (logic analyzers, high-precision oscilloscopes, or hardware timer peripherals) is necessary to assess the impact of timing quantization on the measurements.
6- Questionable SRAM measurement methodology: SRAM usage is measured using an approach based on __heap_start, __brkval, and the address of a local variable (Algorithm 10). The organization of memory, compiler optimizations, interrupt stacks, and library behaviors can alter the meaning of __brkval and heap/stack addresses. A blind subtraction without validation risks systematic under- or over-estimation of SRAM usage. The authors must validate this method (for example, by comparison against linker map reports, toolchain-provided memory usage reports, or instrumented test cases) and discuss limitations.
7- Limited hardware selection reduces industrial relevance: The study focuses exclusively on three 8-bit Arduino boards (Uno, Leonardo, Mega2560). Modern embedded products increasingly employ 32-bit microcontrollers (e.g., ARM Cortex-M series) and hardware accelerators. Results obtained on 8-bit Arduino boards do not necessarily generalize to production-class platforms or streaming/windowed workloads. If the authors want to make claims relevant to industry, experiments must include representative 32-bit MCUs and workload patterns typical of real systems; otherwise, the scope and claims should be explicitly limited to the tested class of devices.
8- Deficient literature review: The state of the art is underdeveloped (only nine references). The manuscript does not situate its contribution adequately relative to prior experimental studies of algorithm performance on constrained platforms, nor does it reference key work on measurement methodology, embedded benchmarking, or memory/time measurement techniques. This limited review weakens the manuscript’s novelty and contextualization.

Reviewer 2 Report

Comments and Suggestions for Authors

General comments:

The topic proposed by the authors is interesting, but its potential has not been fully utilized. The discussed issues regarding the comparison of sorting algorithm performance mainly depend on the computational complexity of the algorithms and the number of data in the set, rather than the amount of available memory in the microcontroller, and the result of such a comparison does not require hardware testing. The memory speed of all three analyzed microcontrollers is identical. The proposed hardware solutions in the form of Arduino Uno, Arduino Leonardo, and Arduino Mega 2560 are anyway outdated. Moreover, the authors stated that “closely resemble real-world use cases” (lines 333-335) refers to integer cases, which is completely incorrect. Modern DSP (Digital Signal Processing) devices, including microcontrollers (eg. with Cortex-M4 ,M7) can  primarily operate on floating-point data. Therefore, the generalized title of the article may be misleading, suggesting to the reader the use of such solutions. The introduction of the article, including the related works section, is very poor. It does not adequately cover the topic proposed by the authors and does not reflect the current state of knowledge in this field, as it was prepared very superficially. References need to be significantly expanded. The article requires a number of changes to be suitable for publication.

Detailed comments requiring answers and changes in the text:

  1. The title needs to be changed; the current form is too general. It should specify that it concerns unoptimized sorting algorithms. Avoid referring to the general concept of “microcontrollers,” as this may mislead readers into thinking of floating-point DSPs. The authors focused only on basic ATmega-family microcontrollers.
  2. References should be ordered according to their appearance in the text.
  3. Section 1.3 refers to IoT and edge devices, but the microcontrollers proposed by the authors in the experiments are not IoT devices per se. The Arduino family is mostly used for control or data collection rather than processing. This section should be supplemented accordingly.
  4. Section 1.4 is very poor. The related work is based on only three publications, which moreover share a common author. The authors did not reference other articles in reputable journals regarding sorting algorithm performance on microcontrollers. This section should be significantly expanded, presenting a wide spectrum of works related to microcontroller performance, including more advanced floating-point DSP solutions, and the literature should also cover topics discussed in the article but currently lacking references.
  5. Why were the optimized algorithms discussed by the authors in Section 1.4 Related Work not included in the experimental part?
  6. A description of the main contributions should appear at the end of Section 1, not in the conclusion.
  7. Due to the article's extended structure, Section 1 at the end should include a brief overview of the contents of the following sections.
  8. Section 2.1 contains many unnecessary details such as board dimensions, connector types, communication interfaces, and pinouts, which are irrelevant to the study. However, there is no information that Arduino boards operate with fixed-point arithmetic.
  9. Section 2.1 should include a table showing memory speed and access times during read and write operations, which is critical for comparing algorithm performance on these devices. It seems that the memory speed of all three analyzed microcontrollers is identical.
  10. Section 2.2 is difficult to read. Algorithm descriptions start with bold text naming the algorithm. It would be better to introduce sub-sections, e.g., 2.2.1 Bubble Sort, etc.
  11. Section 2.2 lacks references and a summary table comparing computational complexity as well as memory requirements for each algorithm described .
  12. Section 2.3 lacks literature references regarding programming language, environment, hardware, and algorithms.
  13. In Section 2.3, the hardware description focuses only on memory size, not speed, which is more significant for algorithm performance.
  14. Why did the authors decide to measure time using the less precise millis() function? The micros() function provides much higher precision and overflows after about 70 minutes, which seems sufficient for the proposed experiments. Using micros() would yield more accurate results. I suggest repeating the timing measurements.
  15. Lines 333-335 state, “Integers were chosen because they closely resemble real-world use cases, since sorting algorithms are typically applied to numeric data.” What is the source of this information? Most data used by DSP devices is floating-point, not integer.
  16. Section 3 requires significant reorganization for clarity. It should start with the results of individual algorithms on specific microcontrollers, and only then present generalized results with averages across all microcontrollers, not the other way around.
  17. Section 3 should start with the current Section 3.5 to show results for individual microcontrollers, which are then averaged. Timing results should be presented in one large table rather than split across six smaller ones (Tables 7, 8, 9, 10, 12, 14) to facilitate comparison. The same applies to Tables 11, 13, 15, which should be combined into one summary table.
  18. Section 3.6 can be moved to Section 4, which should be expanded.
  19. The current Section 3.1 should be expanded with sub-sections for each algorithm for clarity, e.g., 3.2.1 Bubble Sort, etc.
  20. The authors themselves stated in lines 485-486 that "no significant difference in execution speed is observed." This can be determined already at the experiment design stage. The only differences result from a lack of memory on some microcontrollers; there are simply no other differences. So what's the point of conducting such experiments?
  21. The axes on Figures 1–27 lack proper labels; for example, the horizontal axis is missing values 100 and 1000.
  22. Since experiments were conducted for N={100, 250, 500, 750, 1000}, why are the results shown as continuous lines? This approximation is unjustified. Results should be displayed as scatter plots (or bar charts for comparison across three different orders of elements) rather than line graphs. As the graphs show averaged values, tolerance should also be indicated for each point.
  23. Data on Figures 3, 6, 7, 8, 9, 11, 12, 18, 19, 20, 21, 22, 23, 24, 25, 26, 27 often overlaps, making them hard to read. It would be better to use different markers in addition to colors.
  24. All figures have duplicated captions. The Excel-generated captions above the graphs are unnecessary; only the figure number and caption below are needed.
  25. Consider reducing the number of figures by combining, e.g., Figures 19, 20, and 21, etc.
  26. The conclusion lacks numerical information from the experiments conducted.

Reviewer 3 Report

Comments and Suggestions for Authors

This reviewed article discusses a very interesting problem: research and comparison of different sorting algorithms for embedded devices. This is very important problem due to the a very small size of internal SRAM memory available in modern microcontrollers.  The authors did a lot: they implement programs for these algorithms and compare them for different platforms. The paper includes a lot of experiments. The authors shows a lot of results which are discussed. Some interesting and useful conclusions are made. But the paper has some drawbacks which should be eliminated.

  1. There are only words in the Abstract. It is necessary to support the words by some numbers. These numbers can be taken from the experimental part of the paper.  
  2. The Introduction should be corrected in the following way. The choice of sorting algorithms to be investigated should be explained. Next, the authors use a lot of abbreviations without their explanation. All explanations should be introduced before using the abbreviations. 
  3. The Introduction is very long. It includes some information which should be replaced into the part where the related work is discussed. So, I propose to reorganize these two parts (introduction and analysis of related work). Moreover, the part of Related work is very short. Many articles and monographs are devoted to this topic. Please, add more references. I did not find the papers of authors in the references. Please, add our papers to show that you are the experts in the field.
  4. The title «Materials and methods» is very general. Please, change it to take into account the specifics of your  research. Also, in this part, it makes sense to add information about the cost of each microcontroller. Why these microcontrollers are chosen? There are hundreds of different models. Please, justify your choice more precisely.
  5. You show the pseudocode for each sorting algorithm. Are these codes yours ar are taken from the literature. If they are yours, please, prove that they are correct. The same should be done to explain the characteristics of these algorithms.When you write about the characteristics of algorithms, you should point the references these data are taken from.
  6. Have you checked the correctness of your programs implementing the sorting algorithms? How it is done.
  7. I think «Overview of Algorithms» is better as a title of subsection 3.1. In this part, please, explain why there is no need in memory for, for example, the method  Buble Sort (and for other methods, too). 
  8. I think that Figure 3 has no sense because the value memory size is always equal to zero. The same is true for Figure 6 and so on.

As for me, the article includes very interesting results. But the article  should be improved taking into account the remarks from this review.

Round 2

Reviewer 1 Report

Comments and Suggestions for Authors

The authors have made all the required modifications.

Reviewer 2 Report

Comments and Suggestions for Authors

I would like to thank the Authors for their extensive responses to my comments and for making changes to the article's text. It is now much clearer in both its format and its content. In my opinion it can be published in its current form.

Reviewer 3 Report

Comments and Suggestions for Authors

I have analyzed the answers of authors. I think the authors have corrected the article according with my remarks. The article is interesting for readers.

I propose to accept the paper in its current form